# Interplay between cohesin and TORC1 links chromosome segregation and gene expression to environmental changes

**Dorian Besson**[1†]**, Sabine Vaur**[1†]**, Stéphanie Vazquez**[1]**, Sylvie Tournier**[2]**, Yannick Gachet**[2]**, Adrien Birot**[1]**, Stéphane Claverol**[3]**, Adèle L Marston**[4]**, Anastasios Damdimopoulos**[5]**, Karl Ekwall**[5]**, Jean-Paul Javerzat**[1]*

[1]CNRS, Université de Bordeaux - Institut de Biochimie et Génétique Cellulaires, Bordeaux, France; [2]MCD, Centre de Biologie Intégrative, Université de Toulouse, CNRS, UPS, Toulouse Cedex, France; [3]University of Bordeaux, Bordeaux Proteome, Bordeaux, France; [4]Centre for Cell Biology, Institute of Cell Biology, University of Edinburgh, Edinburgh, United Kingdom; [5]Department of Medicine, Karolinska Institutet, Stockholm, Sweden

**\*For correspondence:**
jpaul.javerzat@ibgc.cnrs.fr

[†]These authors contributed equally to this work

## eLife Assessment

This **important** study presents a **compelling** link between nutrient signaling and chromosome regulation, demonstrating that reduced activity in a central nutrient-sensing pathway improves chromosome stability and alters gene expression through effects on cohesin. The **convincing** evidence from a combination of genetic, biochemical and cell biological approaches supports a model in which TORC1-dependent phosphorylation of Mis4 and the cohesin subunit Psm1/Smc1 can modulate cohesin loading to enhance faithful chromosome transmission. While the underlying mechanisms and biological importance of this newly described circuit are not yet fully known, the overall body of evidence is strong and supports the main conclusions.

**Abstract** Cohesin is a DNA tethering complex essential for chromosome structure and function. In fission yeast, defects in the cohesin loader Mis4 result in chromosome segregation defects and dysregulated expression of genes near chromosome ends. A genetic screen for suppressors of the thermosensitive growth defect of *mis4-G1487D* identified several hypomorphic mutants of the Target of Rapamycin Complex 1 (TORC1), a conserved kinase that integrates cellular signals to regulate growth and metabolism through substrate-specific phosphorylation. Here, we demonstrate that the TORC1 pathway modulates cohesin functions in chromosome segregation and gene expression. In the context of compromised cohesin loading, the incidence of chromosome segregation defects was modulated by the growth medium in a TORC1-dependent manner. Pharmacological or genetic downregulation of TORC1 activity restored cohesin binding to its chromosomal sites and improved mitotic chromosome segregation. Notably, reduced TORC1 activity also increased cohesin binding and chromosome transmission fidelity in wild-type cells. These results suggest that environmental cues influence chromosome stability via TORC1. Biochemically, TORC1 co-purified with cohesin and reduced TORC1 activity correlated with decreased phosphorylation of specific residues on Mis4 and cohesin. Mutations in cohesin that mimic the non-phosphorylated state mirrored the effects of TORC1 downregulation, showing that TORC1 is part of the network that controls cohesin phosphorylation to modulate its functions. Finally, we show that the functional interaction between TORC1 and Mis4 extends to the regulation of stress-responsive genes. Our findings reveal a TORC1–cohesin link that may facilitate cellular adaptation to environmental changes. Given that TORC1 inhibitors

and calorie restriction extend lifespan in diverse species, this connection raises the intriguing possibility that cohesin-mediated changes in chromosome structure contribute to these effects.

## Introduction

The cohesin complex is a pivotal player in genome structure and function. Its activity is critical for a variety of biological processes, including sister-chromatid cohesion, nuclear division, DNA replication and repair, and gene expression. Cohesin is an ATP-powered molecular machine that is capable of capturing DNA. Intra-chromosomal DNA capture folds the interphase genome into loops, which contributes to gene regulation, particularly during development and cell fate decisions. Conversely, DNA capture in trans provides cohesion between sister chromatids, which is essential for chromosome segregation and repair (*Zheng and Xie, 2019*; *Davidson and Peters, 2021*; *Oldenkamp and Rowland, 2022*). The core cohesin complex consists of two Structural Maintenance of Chromosome proteins, Smc1 and Smc3 (Psm1 and Psm3 in the fission yeast *Schizosaccharomyces pombe*) whose ATPase heads are bridged by a kleisin subunit (Rad21/Scc1) to which a fourth subunit (hSTAG1-2, spPsc3, scScc3) binds. DNA capture by cohesin requires the so-called loading complex NIPBL/MAU2 (spMis4/Ssl3, scScc2/Scc4) which binds to cohesin and DNA. Inactivation of the cohesin loading machinery before S-phase results in the failure of sister-chromatid cohesion establishment and aberrant chromosome segregation during the ensuing mitosis (*Furuya et al., 1998*; *Ciosk et al., 2000*; *Bernard et al., 2006*). Similarly, DNA loop extrusion is dependent on the NIPBL-Cohesin holocomplex (*Davidson et al., 2019*). Chromatin loops are dynamic structures that form and dissolve on a timescale of minutes. In human, the DNA-binding factor CTCF appears to delineate loop domains, often referred to as topologically associated domains (TADs). Dynamically extruding cohesin complexes are thought to facilitate enhancer–promoter scanning in cis. TAD boundaries (insulation) appear regulated as the strengths of CTCF-anchored loop domains are enhanced when mouse embryonic stem cells exit pluripotency. Consistently, mutations in the genes encoding CTCF and cohesin components are linked to human diseases and developmental abnormalities (*Zheng and Xie, 2019*; *Davidson and Peters, 2021*; *Oldenkamp and Rowland, 2022*). The remodelling of genome architecture appears particularly important for appropriate gene expression during cell fate decisions, suggesting that signalling pathways should convey cellular cues to cohesin. In this context, a genetic screen in mammalian cells identified a set of kinases that alter chromosome folding when inactivated (*Park et al., 2023*).

Here, we reveal a connection between fission yeast cohesin, its loader Mis4 and the Target of Rapamycin Complex 1 (TORC1). TORC1 is a master regulator of cell growth and metabolism, highly conserved in eukaryotes. The kinase activity of TORC1 is stimulated by a variety of intra- and extracellular signals, including nutrients, growth factors, hormones and cellular energy levels. Once activated, TORC1 promotes cell growth and metabolism through the phosphorylation of multiple effectors (*González and Hall, 2017*; *Otsubo et al., 2017*). In mammalian species, growth factors and cellular energy stimulate the activity of mTORC1 through the Rheb GTPase, and the inhibition of the tuberous sclerosis complex (TSC), which functions as a GTPase-activating protein for Rheb. In response to amino acid availability, mTORC1 is activated via RAG GTPases in a TSC-independent manner. Under nutrient-rich conditions, TORC1 promotes anabolic processes, such as protein, nucleotide, and lipid synthesis, while inhibiting catabolic processes, such as autophagy.

In fission yeast, Tor2 provides the catalytic activity of TORC1, while Mip1, the Raptor equivalent, participates in substrate recognition (*Morozumi et al., 2021*). The complex contains three additional subunits, the mLST8 orthologue Wat1/Pop3, Toc1, and Tco89 (*Hayashi et al., 2007*). TORC1 plays a crucial role in switching between cell proliferation and differentiation by sensing nitrogen source. When deprived of nitrogen, fission yeast cells arrest in G1, mate, and undergo meiosis. In the absence of a mating partner, G1 cells enter a G0, quiescent state. Like its mammalian counterpart, the Rheb GTPase Rhb1 is an essential activator of TORC1. Upon nitrogen deprivation, TORC1 activity is restrained by the Gtr1–Gtr2 GTPases, the TSC complex, and the Gcn2 kinase (*van Slegtenhorst et al., 2004*; *Chia et al., 2017*; *Fukuda and Shiozaki, 2018*; *Fukuda et al., 2021*). In conditions of abundant nutrients, TORC1 activity is high but nevertheless attenuated by the Gtr1–Gtr2 heterodimer, which is analogous to the mammalian RAG GTPases (*Chia et al., 2017*; *Fukuda and Shiozaki, 2018*). TORC1 is essential for cell growth, and loss of the TORC1 activity results in cell cycle arrest in

G1. Rapamycin only partially inhibits TORC1 activity in *S. pombe* and does not inhibit growth (*Otsubo et al., 2017*).

The link between cohesin and TORC1 emerged from a genetic screen for mutants able to suppress the thermosensitive growth (Ts) defect of *mis4-G1487D*, which is defective for cohesin loading and chromosome segregation at the restrictive temperature (*Birot et al., 2020*). In principle the screen had the potential to identify regulators of Mis4, with the rationale that loss of a negative regulator should upregulate residual Mis4$^{G1487D}$ activity and restore growth at the restrictive temperature. Alongside the cyclin-dependent kinase (CDK) Pef1, a known negative regulator of Mis4 (*Birot et al., 2020*), several mutants of TORC1 were recovered.

In this study, we explore the connection between cohesin and TORC1. Mis4 plays a crucial role in loading cohesin onto chromosomes and establishing sister-chromatid cohesion during the S-phase of the cell cycle. Cohesin loading is reduced in the *mis4-G1487D* mutant at the non-permissive temperature, resulting in defective sister-chromatid cohesion and chromosome segregation defects during mitosis. The downregulation of TORC1 enhanced cohesin binding to Cohesin-Associated Regions (CARs) and reduced chromosome segregation defects. Likewise, the incidence of chromosome segregation defects was modulated by the growth medium in a TORC1-dependent manner. These observations suggested that the TORC1 complex may control cohesin dynamics. Therefore, we investigated the impact of reducing TORC1 activity on wild-type cells. Indeed, the binding of cohesin to CAR increased. Furthermore, this was accompanied by an increase in the fidelity of chromosome transmission during vegetative growth. These findings raise the intriguing possibility that environmental cues could influence the robustness of chromosome segregation processes.

Affinity purification experiments showed that TORC1 and cohesin co-purified from protein extracts. The phosphorylation level of specific residues on cohesin and its loader was reduced in TORC1 mutants. Phospho-mutants recapitulated most of the effects observed when TORC1 was downregulated, arguing that TORC1 signalling contributes to the phosphorylation status of cohesin and thereby to its regulation.

Finally, we asked whether the TORC1–cohesin link may extend to the regulation of gene expression. Previous work from our group had shown that Mis4 plays a role in controlling gene expression in the sub-telomeric regions of chromosomes (*Dheur et al., 2011*). Since TORC1 is involved in environmental adaptation, we performed transcriptome profiling of Mis4 and TORC1 mutants under different culture conditions. The analysis confirmed that Mis4 affects the expression of a small number of genes with a localization bias away from centromeres and close to telomeres. Interestingly, the identity and number of affected genes differed widely between experimental conditions, suggesting a defective adaptive response. Combining all experiments, 338 genes were differentially expressed in *mis4-G1487D* compared to wild-type. Remarkably, almost all of these genes were also regulated by TORC1. In many cases, the genes were deregulated in the same way and without additive effect, suggesting that Mis4 and TORC1 act in the same pathway.

Collectively, the data presented indicate that TORC1 influences the association of cohesin complexes with chromosomes, regulating their functions in chromosome segregation during mitosis and in the transcriptional response to environmental changes.

## Results

### A genetic screen for suppressors of *mis4-G1487D* identified components of TORC1

The G1487D substitution in the cohesin loader Mis4 renders the strain thermosensitive for growth (Ts, *Figure 1A*). At the restrictive temperature the amount of cohesin associated with chromosomes is reduced and cells exhibit defects in chromosome segregation during mitosis (*Birot et al., 2020*). To identify putative regulators of Mis4, we previously carried out a genetic screen for suppressors of the Ts phenotype of *mis4-G1487D* (*Birot et al., 2020*). One of the suppressors identified was Pef1, a CDK whose activity restrains cohesin binding to CARs. Besides *pef1*, the genetic screen identified three other genes: *tor2* (1 allele), *mip1* (5 alleles), and *caa1* (1 allele). The suppressive effect of the *caa1* mutant was modest (*Figure 1—figure supplement 1*). On the other hand, *tor2* and *mip1* mutants clearly suppressed the Ts phenotype of *mis4-G1487D*, at a level similar to that conferred by the deletion of the *pef1* gene (*Figure 1A*, *Figure 1—figure supplement 1*).

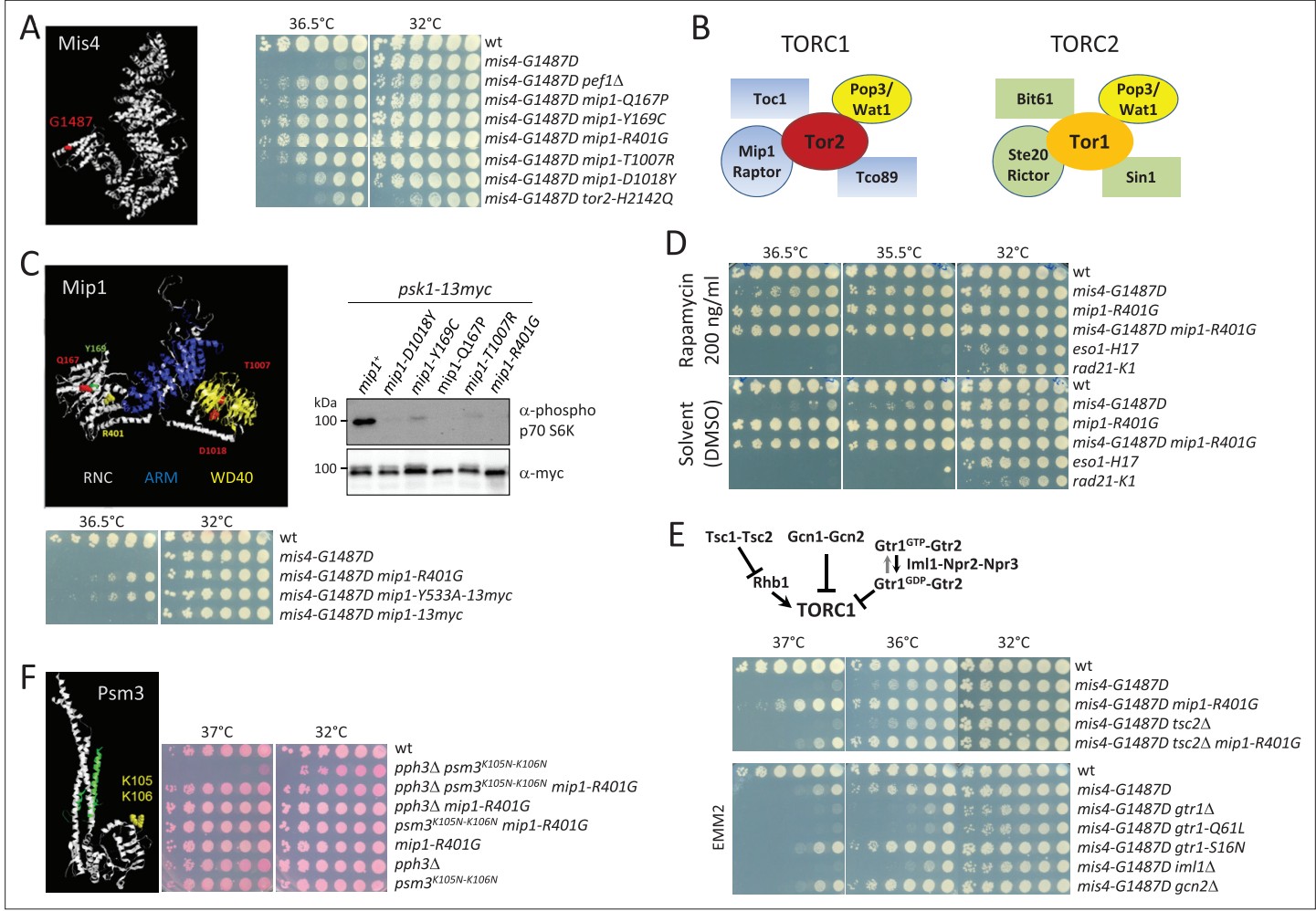

**Figure 1.** Genetic evidence linking TORC1 to cohesin. (**A**) Model structure of Mis4. The G1487D substitution confers a thermosensitive growth (Ts) phenotype. The cell growth assay shows the suppression of the Ts phenotype by the indicated mutations. (**B**) Composition of fission yeast TORC complexes (adapted from *Hayashi et al., 2007*). (**C, D**) Downregulation of TORC1 suppresses the Ts growth defect of *mis4-G1487D*. (**C**) Model structure of Mip1 highlighting amino-acid substitutions in the suppressor mutants (RNC:Raptor N-terminal CASPase-like domain, ARM: Armadillo-type fold, WD40: WD-40 repeats). All *mip1* mutant alleles from the genetic screen showed reduced phosphorylation of the S6-kinase Psk1, a known TORC1 substrate. Conversely, *mip1-Y533A* which is deficient for Psk1 binding and phosphorylation, suppressed the Ts phenotype of *mis4-G1487D*. (**D**) The TORC1 inhibitor rapamycin rescued *mis4-G1487D* but not *eso1* and *rad21* mutants. (**E**) Genetic upregulation of TORC1 exacerbates the Ts phenotype of *mis4-G1487D*. The deletion of *gcn2* or *tsc2* was essentially neutral, although the suppression by *mip1-R401G* was reduced in a *tsc2* deleted background. Deletion of *gtr1*, *iml1*, or *gtr1-Q61L* mimicking the GTP-bound, inactive state of GATOR exacerbated the Ts phenotype of *mis4-G1487D* while mimicking the active GDP-bound form (*gtr1-S16N*) was neutral. (**F**) Psm3$^{K105N-K106N}$ confers a Ts phenotype when combined with the deletion of *pph3*, encoding the catalytic subunit of PP4, a phenotype efficiently rescued by *mip1-R401G*.

The online version of this article includes the following source data and figure supplement(s) for figure 1:

**Source data 1.** Original files for western blot analysis displayed in *Figure 1C*.

**Source data 2.** PDF file containing original western blots for *Figure 1C*, indicating the relevant bands.

**Figure supplement 1.** Genetic interactions between *mis4*, *caa1*, and *tor1*.

The *caa1* gene encodes an aspartate aminotransferase that is required for full activation of the TORC1 complex (*Reidman et al., 2019*). Tor2 and Mip1 are components of TORC1 (*Hayashi et al., 2007*, *Figure 1B*). Both genes are essential for cell survival. The *tor2* allele recovered in our genetic screen bears a substitution in the kinase domain which affected colony growth suggesting that TORC1 activity is largely affected. The five Mip1/Raptor mutants did not show this phenotype, the growth being similar to that of a wild-type strain. Still they all show reduced phosphorylation of the S6-kinase Psk1 (*Figure 1C*), a well-known TORC1 substrate (*Nakashima et al., 2012*). Likewise, the *mip1-Y533A*

mutant which is deficient for Psk1 binding and phosphorylation (*Morozumi et al., 2021*), suppressed the Ts phenotype of *mis4-G1487D* (*Figure 1C*). In a similar manner, the pharmacological downregulation of TORC1 by rapamycin resulted in enhanced growth of *mis4-G1487D* at the restrictive temperature (*Figure 1D*). Other thermosensitive mutants of the cohesin pathway (*eso1-H17* and *rad21-K1*) were not rescued ruling out a general suppressing effect. Hence, the suppression of *mis4-G1487D* may stem from reduced TORC1 kinase activity. Conversely, the genetic upregulation of TORC1 exacerbated the Ts phenotype of *mis4-G1487D*. TORC1 activity upon nitrogen deprivation is restrained by the Gtr1–Gtr2 GTPases, the TSC complex, and the Gcn2 kinase. Among these, the Gtr1–Gtr2 GTPases dampen TORC1 even in nutrient replete conditions (*Figure 1E*, *van Slegtenhorst et al., 2004*; *Chia et al., 2017*; *Fukuda and Shiozaki, 2018*; *Fukuda et al., 2021*). The deletion of *gcn2* or *tsc2* was essentially neutral in terms of *mis4-G1487D* suppression, although the suppression by *mip1-R401G* was reduced in a *tsc2*-deleted background. Deletion of *iml1*, *gtr1* or the *gtr1-Q61L* allele (mimicking the GTP-bound inactive state of the Gtr1 GTPase, *Chia et al., 2017*) exacerbated the Ts phenotype of *mis4-G1487D* while mimicking the active GDP-bound form by using the *gtr1-16N* allele, (*Chia et al., 2017*) was neutral. The deletion of *tor1*, encoding the catalytic subunit of the related complex TORC2 exacerbated the Ts phenotype of the *mis4* mutant (*Figure 1—figure supplement 1*). As TORC1 kinase activity is upregulated when *tor1* is deleted (*Ikai et al., 2011*), this observation is in line with the notion that *mis4-G1487D* is sensitive to hyperactive TORC1.

The above genetic analyses show that the Ts phenotype of the *mis4* mutant is dampened when TORC1 activity is downregulated and exacerbated when TORC1 is upregulated. Because *mip1* alleles barely affected growth on their own and were excellent *mis4* suppressors we focused our analyses on one of them, *mip1-R401G*. We found that *mip1-R401G* acted as suppressor in another genetic context of impaired cohesin loading. The acetyl-mimicking forms of Psm3 have been demonstrated to impede the loading of cohesin (*Murayama and Uhlmann, 2015*; *Hu et al., 2015*; *Birot et al., 2020*). This phenotype is exacerbated by the deletion of *pph3*, encoding the catalytic subunit of Protein Phosphatase 4 and the double mutant strain is Ts for growth (*Birot et al., 2020*). The *mip1-R401G* mutant efficiently suppressed the Ts phenotype of *psm3*$^{K105N-K106N}$ *pph3Δ* (*Figure 1F*), indicating that the suppression is not restricted to *mis4-G1487D* but appears generally related to cohesin loading.

## The occurrence of chromosome segregation defects in *mis4-G1487D* is modulated by TORC1 and growth conditions

When *mis4-G1487D* cells were shifted to the restrictive temperature cell proliferation was reduced and viability dropped ~25-fold after 24 hr when compared to wild-type. The *mip1-R401G* mutation efficiently attenuated the proliferation and viability defects (*Figure 2A*). A hallmark of cohesin mutants is the occurrence of abnormal mitoses in which the spindle apparatus attempts to segregate un-cohered sister chromatids. A single kinetochore is captured by microtubules emanating from opposite spindle poles (merotelic attachment) and moves back and forth on the anaphase spindle as a result of opposite forces (*Pidoux et al., 2000*; *Gregan et al., 2011*). Live cell analysis confirmed the occurrence of merotelic kinetochores in the *mis4* mutant (*Figure 2—figure supplement 1*). In fixed cells, a merotelically attached chromatid appears as DAPI-stained material lagging on the anaphase spindle. Anaphases with lagging DNA were frequent for *mis4-G1487D* after one doubling of the cell population at the restrictive temperature and their occurrence was significantly reduced by *mip1-R401G* (*Figure 2B*). Rapamycin exposure during a single cell cycle also reduced the frequency of anaphases with lagging DNA, although the effect was weaker (*Figure 2C*). This is consistent with the notion that the downregulation of TORC1 reduced chromosome segregation defects of the *mis4* mutant. The suppressor effect of the *mip1-R401G* mutation was not restricted to the *mis4* mutant. The frequency of anaphases with lagging DNA was efficiently reduced in the *psm3*$^{K105N-K106N}$ *pph3Δ* background (*Figure 2D*). As for *mis4-G1487D*, cohesin loading is impaired in this genetic setup (*Birot et al., 2020*) but Mis4 is wild-type. This shows that the suppression is not allele-specific, but may instead stem from an upregulation of the cohesin loading machinery.

Merotelic kinetochores oppose spindle forces and reduce the rate of spindle elongation during anaphase. As the number of merotelic kinetochores increases, the spindle elongation rate slows down and may even become negative resulting in spindle shrinkage (*Courtheoux et al., 2009*). The rate of spindle elongation was strongly affected in *mis4-G1487D* and spindle shrinkage events were frequent, indicative of a high level of merotely. Spindle shrinkage occurred in ~53% of anaphases in *mis4-G1487D*

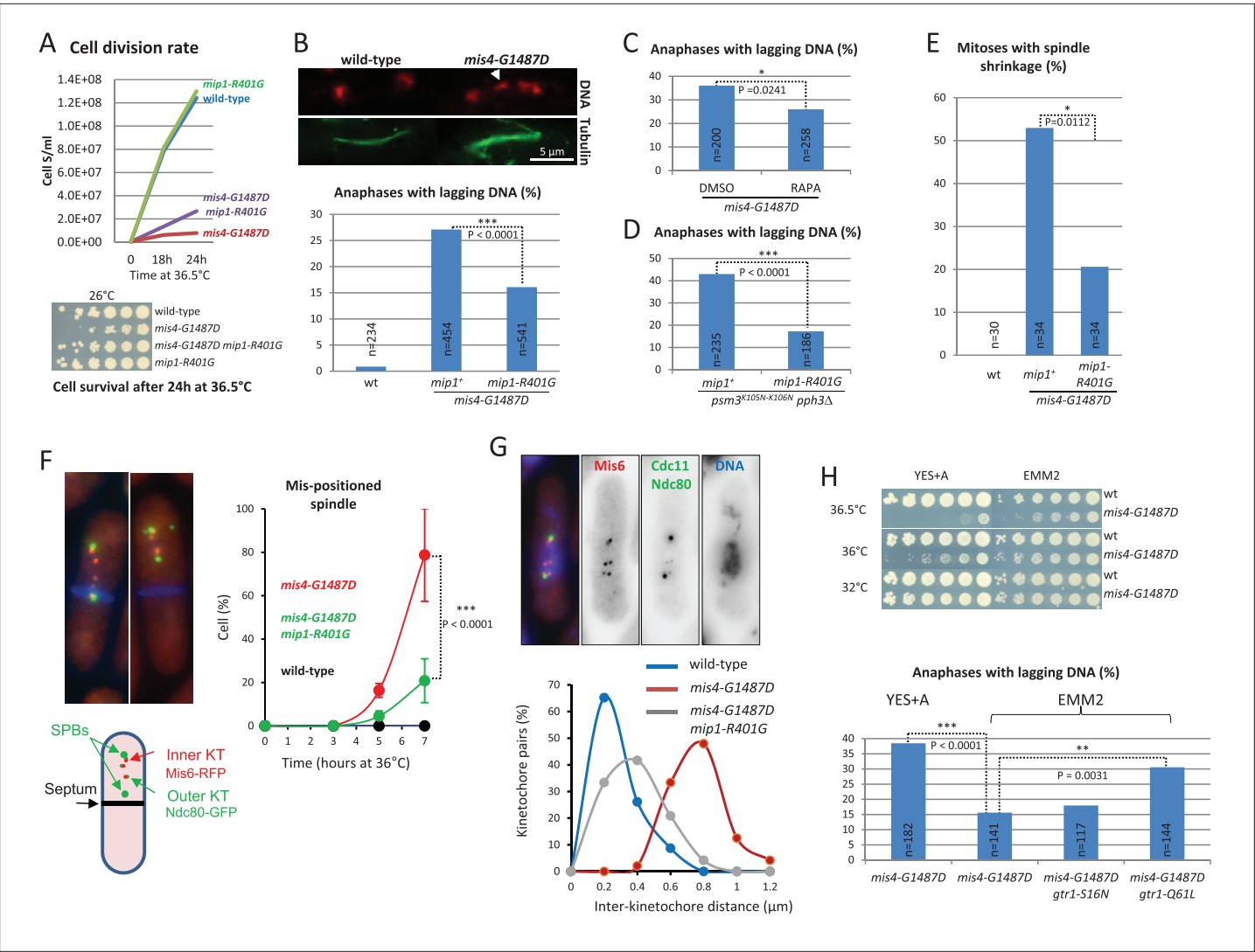

**Figure 2.** The incidence of chromosome segregation defects in *mis4-G1487D* is modulated by TORC1 and the growth medium. (**A**) Exponential growing cells at 25°C were shifted at 36.5°C for 24 hr. The growth curve shows that *mip1-R401G* attenuated the *mis4-G1487D* temperature growth defect. Cell survival after the 24 hr temperature shift was addressed by plating serial fivefold cell dilutions at permissive temperature for *mis4-G1487D* ($4 \times 10^4$ cells in the first row). Cell survival was reduced ~25-fold in *mis4-G1487D* and restored to wild-type levels when combined with *mip1-R401G*. (**B**) Exponentially growing cells at 25°C were shifted at 36°C for one doubling. DNA was stained with DAPI (red, pseudo-colour) and tubulin detected by indirect immunofluorescence (green). Lagging DNA (white arrow) was defined as DAPI-stained material on the anaphase spindle (length >5 µm). (**C**) Rapamycin reduces the incidence of anaphases with lagging DNA. Rapamycin (RAPA, 200 ng/ml) or solvent alone (DMSO) was added to cycling cells at 25°C and the cultures shifted to 36°C for one doubling of the cell population. Samples were treated as in B. (**D**) The *mip1* mutant reduced the frequency of anaphases with lagging DNA in the *psm3^{K105N-K106N} pph3Δ* background. Cells were treated as in B. (**E**) Spindle shrinkage events (shrinkage of more than 0.5 µm) were detected by live analysis of cells undergoing anaphase at 36.5°C (*Figure 2—figure supplement 2*). (**F**) Anaphase cells with displaced spindle. Cells were cultured at 36.5°C, fixed and stained with Calcofluor to visualize septa (blue). Shown are mean and SD from three independent experiments (*n* > 65 cells per strain and per experiment). (**G**) Measurement of inter-kinetochore distance. Cells were shifted to 36.5°C for 5 hr and subsequently chilled at 4°C for 30 min. Cells were fixed, DNA stained with DAPI and the distance between Mis6 signals was measured. The images on the top show scattered kinetochore pairs in the *mis4-G1487D* mutant. Bottom, graph showing the distribution of inter-kinetochore distances (wt, *n* = 46; *mis4-G1487D*, *n* = 48; *mis4-G1487D mip1-R401G*, *n* = 48). (**H**) Top, cell growth assay showing that the Ts growth defect of *mis4-G1487D* is more severe in YES+A medium that in EMM2. Bottom, the frequency of anaphases with lagging DNA is higher in YES+A than in EMM2. The inactive form of the TORC1 inhibitor Gtr1 (Gtr1-Q61L) increased the frequency of abnormal anaphases while the active form was neutral (Gtr1-S16N). Cells were treated as in B. (*) (**) (***), p-values from two-sided Fisher's exact test.

The online version of this article includes the following source data and figure supplement(s) for figure 2:

**Figure supplement 1.** Live analysis of *mis4-G1487D* undergoing anaphase at 36.5°C.

**Figure supplement 2.** Live analysis of anaphase cells revealed spindle shrinkage events.

*Figure 2 continued on next page*

*Figure 2 continued*

**Figure supplement 3.** Sister chromatid cohesion and the level of cohesin acetylation remain low in the *mis4-G1487D mip1-R401G* strain.

**Figure supplement 3—source data 1.** Original files for western blot analysis displayed in panel B.

**Figure supplement 3—source data 2.** PDF file containing original western blots for panel B, indicating the relevant bands.

**Figure supplement 4.** The frequency of anaphases with lagging DNA is dependent on the culture medium.

whereas it was reduced to ~21% in the *mip1-R401G* background (*Figure 2E*, *Figure 2—figure supplement 2*). Particularly striking was the observation of spindle shrinkage followed by asymmetric spindle displacement resulting in the generation of anucleate daughter cells after cytokinesis (*Figure 2F*). Nearly 80% of cells showed this phenotype in *mis4-G1487D* for the late time points whereas the frequency was reduced to ~20% in the presence of *mip1-R401G*. Altogether, these analyses indicate that the *mis4* mutant experienced numerous merotelic events during anaphase at the restrictive temperature, leading to asymmetric spindle shrinkage and the production of anucleate cells. The suppression of these phenotypes by *mip1-R401G* is consistent with improved sister-chromatid cohesion, leading to a lower incidence of merotelic attachments and chromosome segregation defects. At centromeres, defective cohesion results in increased distance between sister-kinetochores, a phenotype that can be evidenced by microscopy. When early mitotic cells are abruptly cooled on ice, microtubules disassemble leaving unattached kinetochore pairs (*Gachet et al., 2008*). In some instances a kinetochore pair drifts away from the other chromosomes, allowing the measurement of inter-kinetochore distance. The inter-kinetochore distance increased in the *mis4* mutant compared to the wild-type. Importantly, the defect was reduced in the *mip1-R401G* background (*Figure 2G*). The suppression was nevertheless partial, suggesting sister-centromere cohesion was not fully restored. Using a GFP-tagged *ade6* locus on the right arm of chromosome III, we found that sister-chromatid cohesion remained defective at that site (*Figure 2—figure supplement 3*). The level of Psm3 acetylation, a marker of sister-chromatid cohesion establishment, was low in the *mis4* mutant and was not restored by *mip1-R401G* or rapamycin. These data suggest that sister-chromatid cohesion remains defective overall, but has improved sufficiently, particularly at the centromeres, to enable more effective chromosome segregation during mitosis.

Physiologically TORC1 acts as a rheostat, its activity rising with nutrient availability. YES is a rich undefined medium containing numerous sources of nitrogen, whereas ammonia is the sole source of nitrogen in EMM2. The Ts phenotype of *mis4-G1487D* was stronger and the frequency of lagging DNA was higher in the rich, complete medium YES than in the synthetic minimal medium EMM2 (*Figure 2H*). The artificial increase of TORC1 activity by *gtr1-Q61L* resulted in an elevated rate of anaphases with lagging DNA in EMM2 medium (*Figure 2H*). Conversely, the use of glutamate, a nitrogen source slightly poorer than ammonia (*Davie et al., 2015*) reduced chromosome segregation defects (*Figure 2—figure supplement 4*). Altogether, these analyses show that the incidence of chromosome segregation defects of *mis4-G1487D* is modulated by growth conditions in a TORC1-dependent manner.

## Reduced TORC1 activity increased Mis4 and cohesin binding to CARs

The cohesin loading complex performs its essential function during G1/S (*Furuya et al., 1998*; *Ciosk et al., 2000*; *Bernard et al., 2006*). The downregulation of TORC1 may rescue *mis4-G1487D* by upregulating its residual cohesin loading activity. To address this issue, cells were arrested at the end of the G1-phase by the use of the *cdc10-129* Ts mutation and cohesin binding to chromosomes monitored by Rad21 chromatin immunoprecipitation (ChIP)-qPCR at known CARs (*Figure 3A*). Cycling *S. pombe* cells are essentially in the G2-phase of the cell cycle (*Carlson et al., 1999*). When shifted at 36.5°C, *cdc10-129* cells progress through G2, M, and arrest in G1. As reported previously, Rad21 binding to CARs was reduced in a *mis4-G1487D* background (*Feytout et al., 2011*; *Vaur et al., 2012*; *Birot et al., 2020*). The *mip1-R401G* mutation efficiently restored cohesin binding to near wild-type levels (*Figure 3B*). A similar effect was observed in the *psm3^{K105N-K106N} pph3Δ* background (*Figure 3—figure supplement 1*) indicating that *mip1-R401G* restores cohesin loading in these two distinct genetic contexts. Importantly, a similar stimulating effect was observed in cells with a wild-type cohesin loading machinery. In an otherwise wild-type background, Mis4 and Rad21 binding to CARs were increased in *mip1-R401G* cells at most chromosomal sites examined (*Figure 3C, D*).

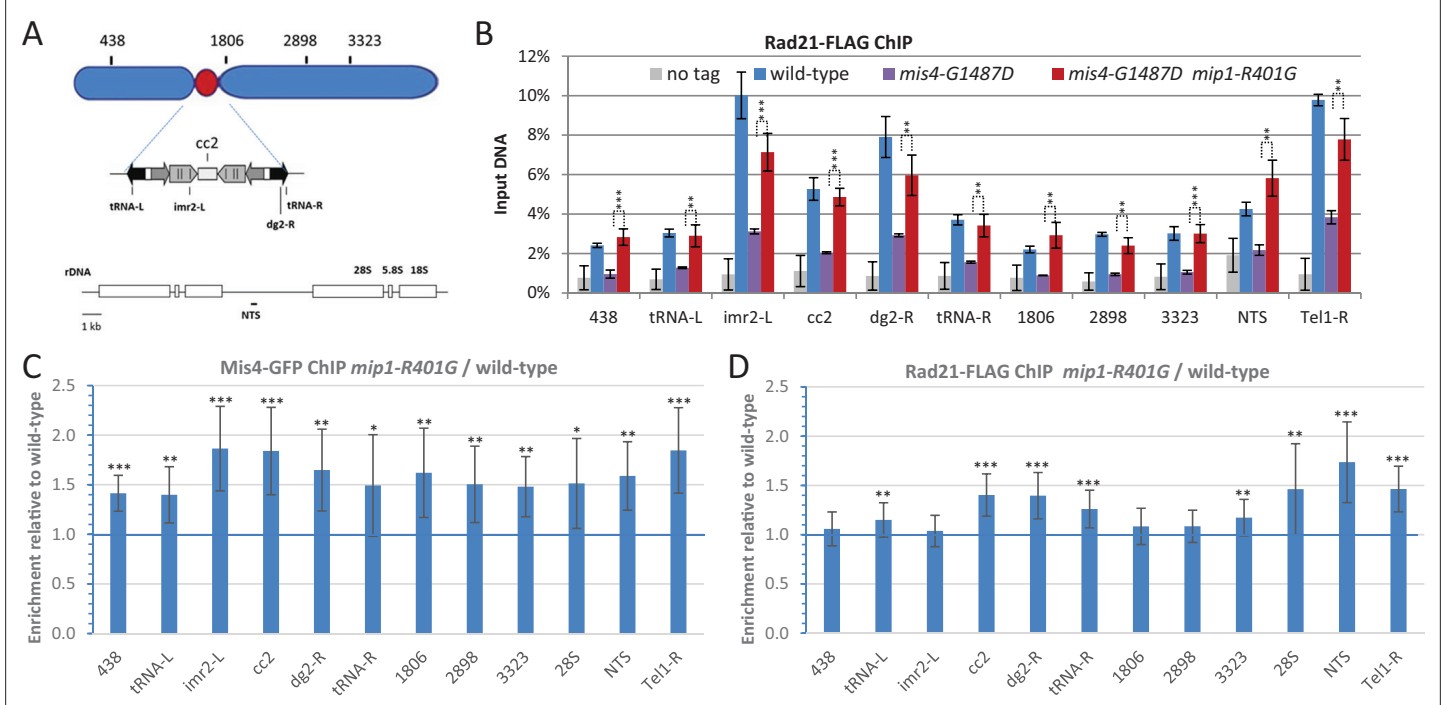

**Figure 3.** Increased cohesin binding to Cohesin-Associated Regions in the raptor mutant *mip1-R401G*. (**A**) Chromosomal sites examined were Cohesin-Associated Regions along the arms (438, 1806, 2898, and 3323) and centromere of chromosome 2, the non-transcribed spacer region (NTS) of the rDNA gene cluster on chromosome 3 and the chromosome 1 right telomere. Within the centromere, the central core (cc2) which is the site of kinetochore assembly, the imr and dg repeats that flank the central core on either side and at tRNA-rich domains that delineate the centromere. (**B**) The *mip1-R401G* mutation restores cohesin binding in *mis4-G1487D* cells. Cohesin binding to chromatin was monitored by Rad21-FLAG ChIP in G1 cells (*cdc10-129* arrest) at 36.5°C. Error bars = SD from 4 ChIPs (technical replicates), except for *mis4-G1487D* (3 ChIPs). \*\*\*p ≤ 0.001, \*\*p ≤ 0.01, \*p ≤ 0.05 by two-tailed, unpaired *t*-test with 95% confidence interval. (**C, D**) The *mip1-R401G* mutation increased Mis4 and Rad21 binding in *mis4*⁺ cells. ChIP assays were made with *cdc10-129*-arrested cells. (**C**) Mean ratios ± SD were calculated from 2 independent experiments and 4 technical replicates per experiment. (**D**) Mean ratios ± SD were calculated from 4 independent experiments and 4 technical replicates per experiment. \*\*\*p ≤ 0.001, \*\*p ≤ 0.01, \*p ≤ 0.05, by two-tailed, one sample *t*-test with 95% confidence interval.

The online version of this article includes the following figure supplement(s) for figure 3:

**Figure supplement 1.** Genetic or pharmacological downregulation of TORC1 increases the binding of cohesins to CARs under wild-type conditions or when the cohesin loading machinery is impaired.

Rapamycin treatment during a single cell cycle provoked a similar stimulation of Rad21 binding at CARs (*Figure 3—figure supplement 1*), albeit with noticeable differences. In *mis4*⁺ cells, both *mip1-R401G* and rapamycin induced a significant increase in Rad21 binding at several CARs (*tRNA-left*, *cc2*, *3323*, *NTS*, *Tel1-R*). However, some CARs that exhibited increased Rad21 binding in the *mip1* mutant did not respond significantly to rapamycin (*dg2-R*, *tRNA-R*). Conversely, rapamycin (but not *mip1-R401G*) induced a significant increase in Rad21 binding at *imr2-L* and *CAR1806* (*Figure 3D*, *Figure 3— figure supplement 1*). In the *mis4-G1487D* mutant background, *mip1-R401G* induced a significant increase in Rad21 binding at all examined sites (*Figure 3B*). Similarly, rapamycin did increase Rad21 binding at all sites but only at the *Tel1-R* site did this reach statistical significance (*Figure 3—figure supplement 1*). We conclude that TORC1 downregulation results in increased cohesin abundance at CARs, presumably by stimulating cohesin loading.

It is important to note at this stage that, although rapamycin and TORC1 mutants both decrease TORC1 kinase activity, the two are not equivalent. The mechanisms by which TORC1 kinase activity is reduced are different, and TORC1 mutants suppress the *mis4-G1487D* phenotype more effectively than rapamycin. It is known that bona fide TORC1 substrates respond differently to rapamycin. Some phosphosites show acute sensitivity, while others are less sensitive or even insensitive (*Kang et al., 2013*). TORC1 mutants cause a constitutive decrease in activity with possible adaptive effects, whereas rapamycin is applied for a single cell cycle. While the lists of affected TORC1 substrates may overlap, they are unlikely to be identical. Furthermore, the phosphorylation level of the relevant substrates

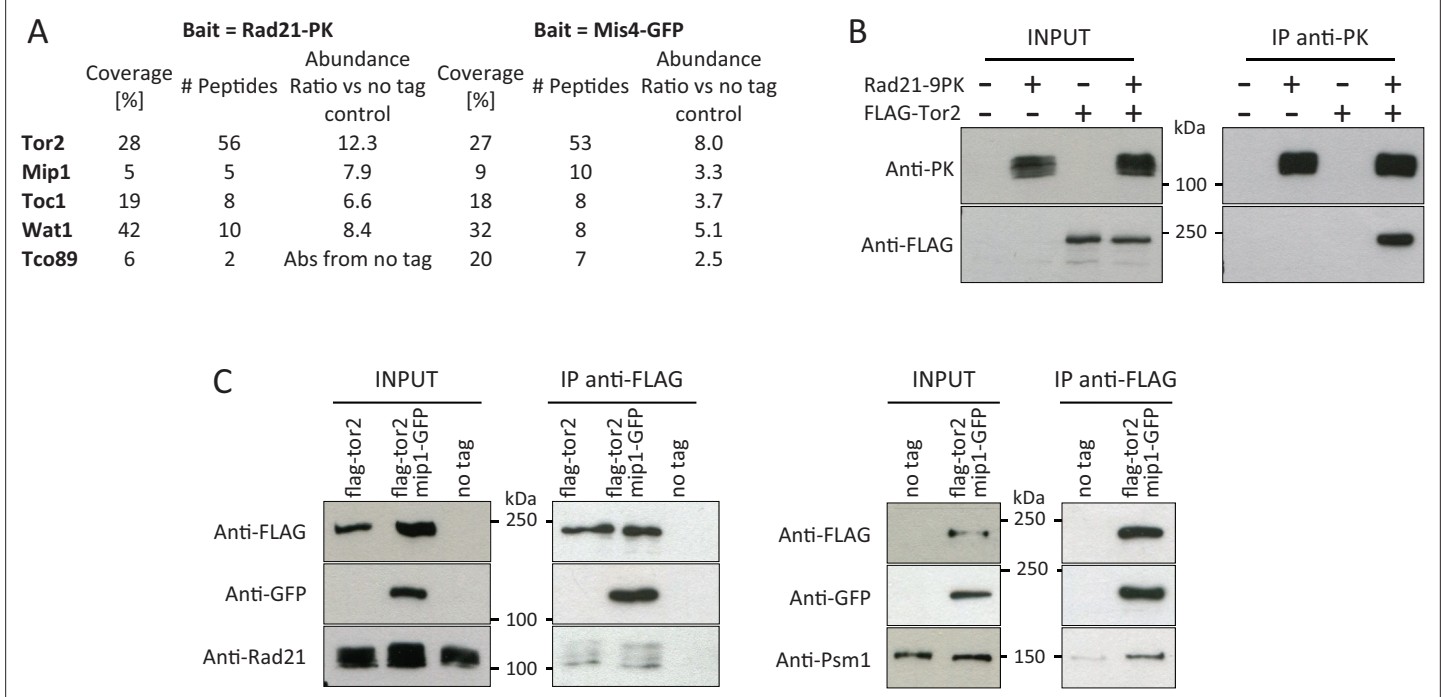

**Figure 4.** TORC1 components co-purify with Rad21 and Mis4. (**A**) Affinity purifications (triplicate samples) were made from G1-arrested cells (*cdc10-129* arrest) and proteins analysed by label-free mass spectrometry. (**B, C**) Reciprocal co-immunoprecipitation experiments. Western blots were probed with the indicated antibodies. (**B**) Rad21-PK IP (*cdc10-129*-arrested cells). (**C**) Tor2-FLAG IP (cycling cells).

The online version of this article includes the following source data and figure supplement(s) for figure 4:

**Source data 1.** Original files for western blot analysis displayed in *Figure 4B, C*.

**Source data 2.** PDF file containing original western blots for *Figure 4B, C*, indicating the relevant bands.

**Figure supplement 1.** The *mip1-R401G* mutation does not prevent the Tor2 kinase from co-immunoprecipitating with Mis4 and Rad21.

**Figure supplement 1—source data 1.** Original files for western blot analysis displayed in panels B and C.

**Figure supplement 1—source data 2.** PDF file containing original western blots for panels B and C, indicating the relevant bands.

is not necessarily altered to the same extent. It is therefore remarkable that negative regulation of TORC1 by rapamycin or a genetic mutation both alleviate *mis4-G1487D* phenotypes and have a fairly similar effect on cohesin dynamics.

## TORC1 components co-purify with cohesin and its loader

To see whether TORC1 and cohesin may interact we affinity-purified Mis4 and Rad21 complexes and analysed associated proteins by label-free mass spectrometry (*Figure 4A*). To avoid cell cycle effects, extracts were prepared from G1 (*cdc10-129*)-arrested cells. All TORC1 components co-purified with Mis4 and Rad21 and conversely, additional co-immunoprecipitation experiments showed that cohesin co-purified with TORC1 (*Figure 4B, C*). Tor2, the kinase subunit of TORC1, is particularly well detected in Rad21 and Mis4 co-immunoprecipitation experiments (*Figure 4*, *Figure 4—figure supplement 1*). To determine whether the R401G mutation in Mip1 affects these interactions, co-immunoprecipitation experiments were repeated in both the *mip1-R401G* and *mip1*[+] contexts. The data obtained indicate that Tor2 co-immunoprecipitation with Mis4 and Rad21 is largely unaffected by the *mip1-R401G* mutation (*Figure 4—figure supplement 1*). If *mip1-R401G* affects the regulation of cohesin by TORC1, this does not appear to stem from a gross defect in their interaction, at least at this level of resolution.

## TORC1 signalling affects the phosphorylation levels of Mis4-S183 and Psm1-S1022

To see whether TORC1 activity may affect cohesin or Mis4 phosphorylation status, Mis4 and Rad21 complexes were purified from *mip1-R401G* and *mip1+* cells and triplicate samples analysed by label-free mass spectrometry. As before, protein extracts were prepared from G1-arrested cells to avoid cell cycle induced changes. A serine residue (S183) within the N terminal domain of Mis4 was less phosphorylated in *mip1-R401G* compared to wild-type. Likewise, Psm1-S1022 phosphorylation level was decreased (*Figure 5A*). Antibodies were raised against the phosphorylated forms (*Figure 5—figure supplements 1 and 2*) and this confirmed that the levels of Mis4-S183p and Psm1-S1022p were reduced in *mip1-R401G* cells (*Figure 5B, C*). A similar reduction was also apparent in the unrelated mutant *mip1-Y533A* (*Figure 5—figure supplement 1*). Rapamycin exposure during one complete cell cycle led to a reduction of Psm1-S1022 phosphorylation (*Figure 5D*). Mis4-S183 behaved differently as its phosphorylation was increased by rapamycin (*Figure 5E*). Altogether, these data show that the phosphorylation levels Mis4-S183 and Psm1-S1022 are responsive to alterations in TORC1-signalling.

Both phosphorylation sites on Psm1 and Mis4 adhere to the CDK consensus (S–T/P) but were not described as direct targets of Cdc2, the main fission yeast CDK (*Swaffer et al., 2018*). Label-free mass spectrometry indicated that the phosphorylation of Mis4-S183 was not dependent on the CDK Pef1 either (*Figure 5—figure supplement 2*). However, Psm1-S1022 phosphorylation was strongly reduced in *pef1* deleted cells (*Figure 5—figure supplement 2*). Pef1 was reported as a positive regulator of TORC1 (*Matsuda et al., 2020*), suggesting Pef1 may promote the phosphorylation of Psm1 through TORC1 activation. Indeed, the phosphorylation level of the TORC1 substrate Psk1 was reduced in *pef1* deleted cells and the double mutant *pef1Δ mip1-R401G* grew very poorly (*Figure 5—figure supplement 2*), consistent with further reduced TORC1 activity. The phosphorylation of Psm1 is therefore possibly dependent on the activation of TORC1 by Pef1. Conversely, Pef1 activity does not seem to depend on TORC1 as the phosphorylation status of Rad21-T262, which is a Pef1 substrate (*Birot et al., 2020*), was not affected by *mip1-R401G* (*Figure 5—figure supplement 2*).

TORC1 may regulate cohesin phosphorylation directly, or via the activation of downstream kinases. In fission yeast there are three known AGC kinases phosphorylated and thereby presumably activated by TORC1: Psk1, Sck1, and Sck2 (*Nakashima et al., 2012*; *Chica et al., 2016*). The ribosomal S6 kinase Psk1 was introduced earlier. The closest homologs of Sck1–2 are SGK1–3 (Serum/glucocorticoid regulated kinase 1, 2, and 3) in human and SCH9 in budding yeast. AGC kinase genes are dispensable for growth in fission yeast, either alone or in combination, which rendered possible a straightforward analysis.

Deletion of the *psk1* gene was essentially neutral for *mis4-G1487D* (*Figure 5—figure supplement 3*). The deletion of *sck1* had a weak suppressor effect whereas *sck2* deletion suppressed the growth defect to a level similar to *mip1-R401G*. The double deletion of *sck1* and *sck2* had the same effect as the single deletion of *sck2*. Using Rad21-FLAG ChIP-qPCR on G1-arrested cells we determined the relative enrichment of cohesin localized at CARs in the mutant strains versus wild-type (*Figure 5—figure supplement 3*). A small decrease was observed for *sck1Δ* at several CARs, such as 438, tRNA-L, cc2, imr2-L, and 1806. The *sck2Δ* single mutant behaved quite similar to wild-type, at least in G1-arrested cells. In contrast, the *sck1Δ sck2Δ* double mutant showed an increase relative to wild-type at almost all loci examined and at a level similar to *mip1-R401G*. This indicates a synthetic effect of the simultaneous deletion of *sck1* and *sck2* on cohesin association at CARs in G1-arrested cells. The frequency of chromosome segregation defects in the *mis4-G1487D* strain remained unchanged in a *sck1*-deleted background, but was significantly reduced when either the *sck2* or both the *sck1* and *sck2* genes were deleted (*Figure 5—figure supplement 3*).

The phosphorylation of Psm1-S1022 was close to the wild-type level in the *sck1Δ* and *sck2Δ* single mutants (76% and 87%, respectively). The *sck1Δ sck2Δ* double mutant showed a 45% decrease in phosphorylation (*Figure 5—figure supplement 3*). These results suggest that *sck1* and *sck2* are redundant for the regulation of Psm1-S1022 phosphorylation. The phosphorylation level of Mis4-S183 was increased 5.6-fold in *sck1Δ* compared to wild-type. Conversely, Mis4-S183p was reduced in the *sck2Δ* strain (50% decrease relative to wild-type) and the double mutant showed a fivefold higher signal than the wild-type (*Figure 5—figure supplement 3*). As Mis4-S183 is phosphorylated in the double mutant this indicates that Mis4-S183 is phosphorylated by another, unknown kinase, at least in

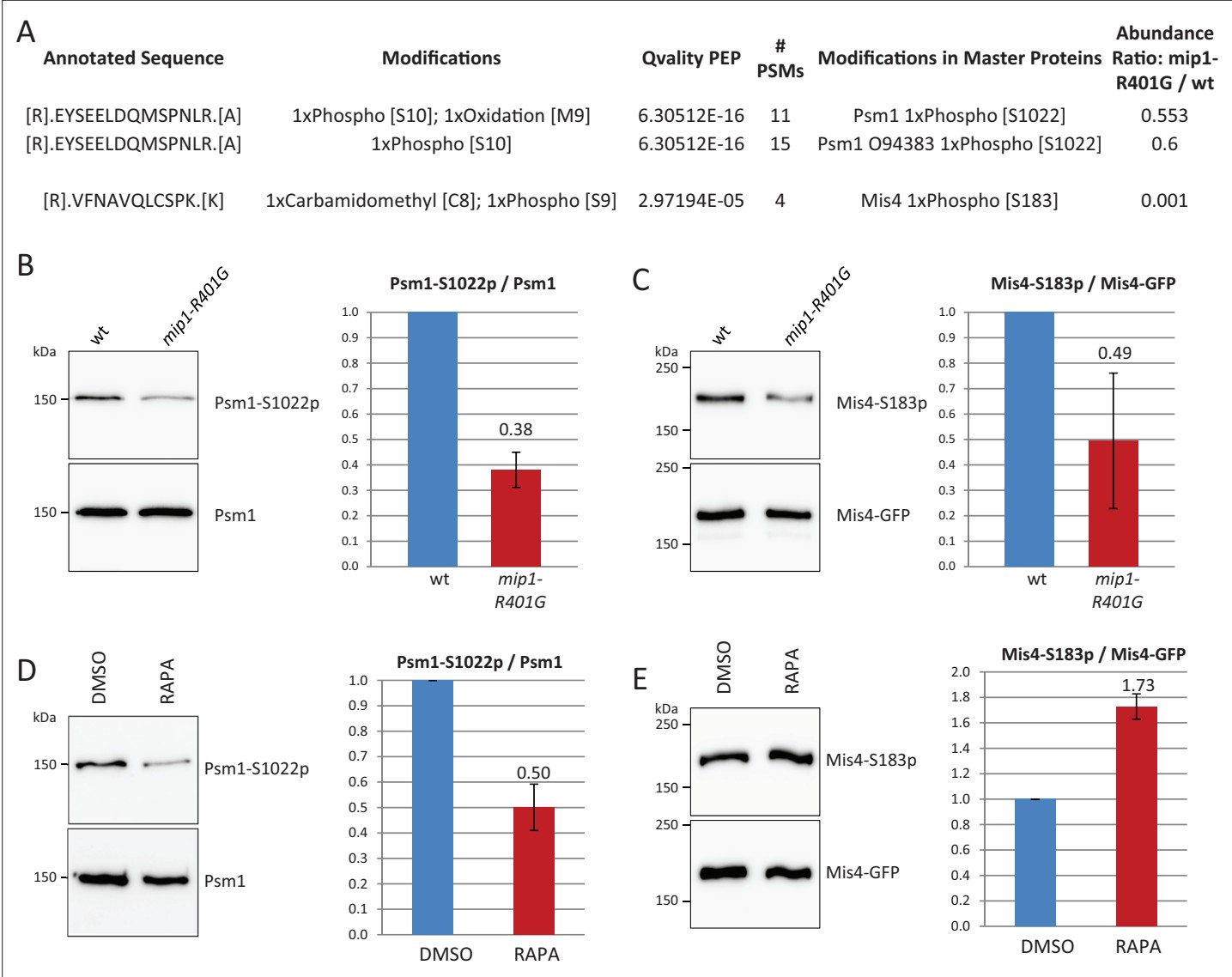

**Figure 5.** TORC1 signalling affects the phosphorylation levels of Psm1-S1022 and Mis4-S183. (**A**) Mis4-GFP and Cohesin (Rad21-PK) were affinity-purified from G1-arrested cells (*cdc10-129* arrest). Triplicate samples were analysed by label-free mass spectrometry. Mis4-S183 and Psm1-S1022 phosphorylation levels were reduced in the *mip1-R401G* background. (**B**) Cohesin was purified by Rad21-FLAG immunoprecipitation from *cdc10-129*-arrested cells and probed with anti-Psm1-S1022p and anti-Psm1 antibodies. Mean ratios ± SD from three independent experiments. (**C**) Mis4-GFP was affinity-purified from *cdc10-129*-arrested cells and probed with anti-Mis4-S183p and anti-GFP antibodies. Mean ratios ± SD from three independent experiments. (**D, E**) Effect of rapamycin on the phosphorylation status of Psm1-S1022 and Mis4-S183. Rapamycin (RAPA, 200 ng/ml) or solvent alone (DMSO) was added to cycling cells at 25°C and the cultures shifted to 36°C for one doubling of the cell population. (**D**) Cohesin was purified by Rad21-FLAG immunoprecipitation and probed with anti-Psm1-S1022p and anti-Psm1 antibodies. Mean ratios ± SD from two technical replicates. (**E**) Mis4-GFP was affinity purified and probed with anti-Mis4-S183p and anti-GFP antibodies. Mean ratios ± SD from two technical replicates.

The online version of this article includes the following source data and figure supplement(s) for figure 5:

**Source data 1.** Original files for western blot analysis displayed in panels B–E.

**Source data 2.** PDF file containing original western blots, indicating the relevant bands for panels B–E.

**Figure supplement 1.** Reduced phosphorylation levels of Mis4-S183 and Psm1-S1022 in the Raptor mutant *mip1-Y533A*.

**Figure supplement 1—source data 1.** Original files for western blot analysis displayed in panels A–C.

**Figure supplement 1—source data 2.** PDF file containing original western blots, indicating the relevant bands for panels A–C.

**Figure supplement 2.** Psm1-S1022 phosphorylation depends on Pef1 CDK, whereas Mis4-S183 does not.

**Figure supplement 2—source data 1.** Original files for western blot analysis displayed in panels B, D, and E.

*Figure 5 continued on next page*

*Figure 5 continued*

**Figure supplement 2—source data 2.** PDF file containing original western blots, indicating the relevant bands for panels B, D, and E.

**Figure supplement 3.** Regulation of cohesin by AGC kinases downstream of TORC1.

**Figure supplement 3—source data 1.** Original files for western blot analysis displayed in panels D and E.

**Figure supplement 3—source data 2.** PDF file containing original western blots, indicating the relevant bands for panels D and E.

this genetic context. In this scenario, the unknown kinase would be positively regulated by Sck2 and negatively by Sck1.

The consensus site for Sck1 and Sck2 is unknown. If we assume some conservation with budding yeast SCH9, the consensus sequence would be RRxS/T. Psm1-S1022 (DQMSP) and Mis4-S183 (QLCSP) do not fit the consensus. However, this should be taken with care as many SCH9-dependent phosphorylation sites did not fall within the consensus in a study using analogue-sensitive AGC kinases and phosphoproteomics (*Plank et al., 2020*). Alternatively, Sck1–2 may regulate other kinases. Indeed Psm1-S1022 and Mis4-183 lie within CDK consensus sites and Psm1-S1022 phosphorylation is Pef1-dependent.

The interplay between TORC1, Pef1 and other kinases downstream of TORC1 appears complex and remains to be elucidated. Nevertheless, the above data show that TORC1 signalling affects the phosphorylation level of Mis4 and Psm1, the abundance of cohesin at CARs and the suppression of *mis4-G1487D* phenotypes.

## Mis4-S183 and Psm1-S1022 are relevant targets for the regulation of cohesin by TORC1

The *mip1-R401G* mutant is a strong suppressor of *mis4-G1487D* phenotypes and the phosphorylation levels of Mis4-S183 and Psm1-S1022 are reduced. To address the causality of the phosphorylation status, we generated mutants mimicking the non-phosphorylated (S to A) or phosphorylated state (S to E/D). Non-phosphorylatable *mis4* and *psm1* mutants rescued *mis4-G1487D* growth and chromosome segregation defects while phospho-mimetics had the opposite effect (*Figure 6A–D*). Cohesin binding to CARs as assayed by ChIP tend to increase for the mutants mimicking the non-phosphorylated state and to decrease with the phospho-mimicking forms (*Figure 6E*). The rescue of *mis4-G1487D* by the non-phosphorylatable form was modest but significant, notably within centromeric regions (*imr2-L*, *dg2-R*) and at the telomere (*Tel1-R*) site (*Figure 6E* and see *Figure 6—figure supplement 1* for comparison with wild-type levels). Conversely, the mutant mimicking the phosphorylated state displayed a significant reduction of Rad21 binding at those sites as well as to several other sites at the centromere (*cc2*, *tRNA-R*), *CAR2898*, and at the ribosomal non-transcribed spacer site *NTS*. Likewise, non-phosphorylatable *mis4* and *psm1* mutants efficiently rescued growth and chromosome segregation defects in the *psm3$^{K105N-K106N}$ pph3Δ* background and ChIP analyses revealed that cohesin binding to CARs was clearly stimulated (*Figure 6—figure supplement 2*). These data show that the phenotypes of *mip1-R401G* are largely recapitulated by mutants mimicking the non-phosphorylated state, arguing that Mis4 and Psm1 are relevant downstream effectors for cohesin regulation by the TOR signalling pathway.

## Rapamycin increased the fidelity of chromosome transmission of wild-type cells in a pathway involving Psm1 phosphorylation

The data we have obtained so far show that reducing TORC1 activity reduced the severity of chromosome segregation defects in genetic contexts where cohesin loading is compromised. We therefore wondered whether the alteration of TORC1 activity could affect the fidelity of chromosome transmission in the wild-type. The loss frequency of whole, natural chromosomes can be estimated by measuring the breakdown of diploids (*Allshire et al., 1995*). Remarkably, diploids were more stable when grown in the presence of rapamycin when compared to solvent alone (*Figure 7A*). We also measured the loss frequency of CM3112, a circular, non-essential minichromosome derived from the centromere region of chromosome III (*Matsumoto et al., 1990*). In the wild-type, it is lost at a rate of around 1 per 100 divisions. Again, rapamycin led to a substantial decrease in the incidence of CM3112 loss (*Figure 7B*). The frequency of CM3112 loss was reduced in the *mip1-R401G* mutant when compared to wild-type. Furthermore, the combination of rapamycin and *mip1-R401G* did not

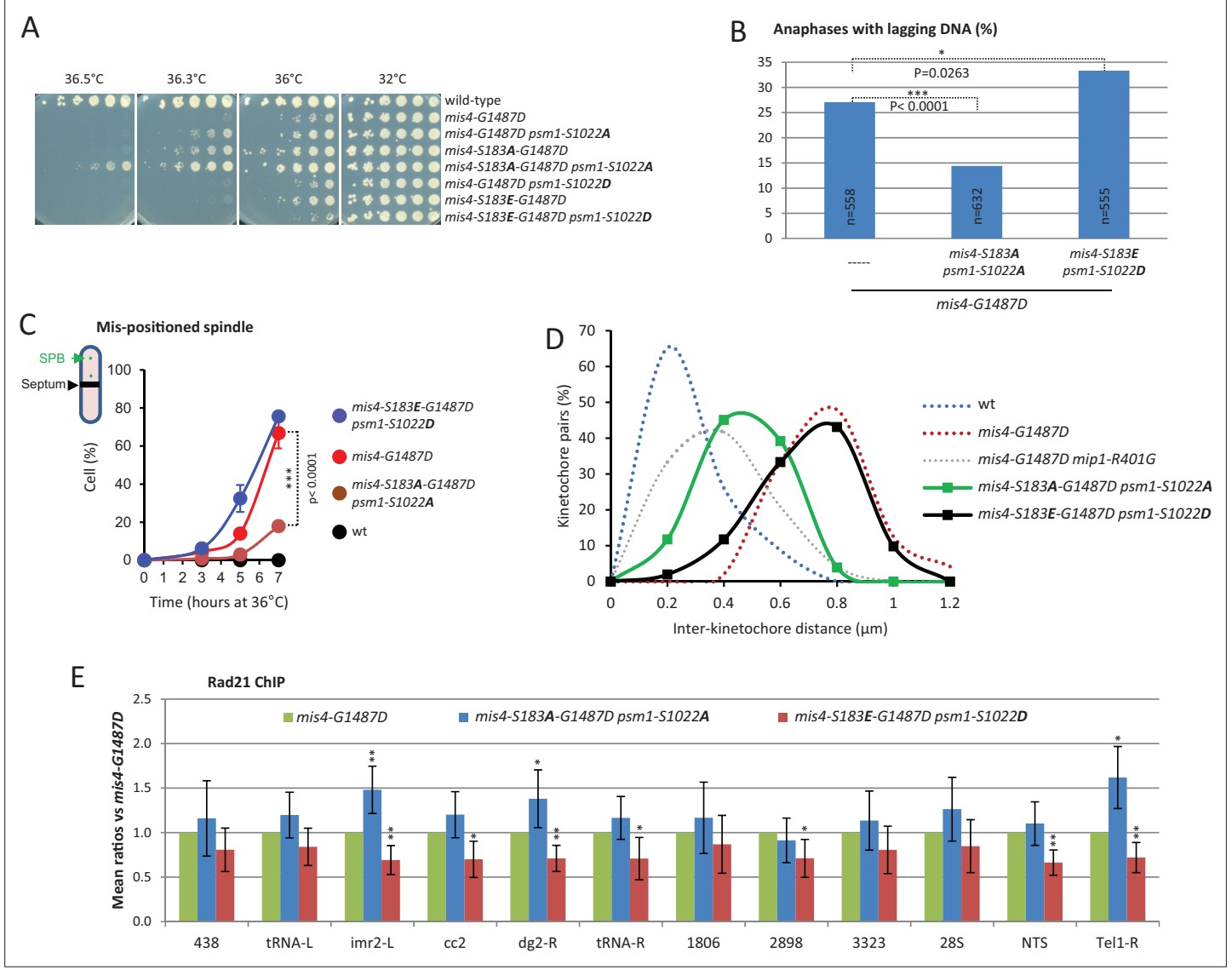

**Figure 6.** Mimicking the non phosphorylated state of Psm1-S1022 and Mis4-S183 alleviates *mis4-G1487D* phenotypes. (**A**) Cell growth assays. (**B**) Anaphases with lagging DNA. Exponentially growing cells at 25°C were shifted at 36°C for 3.5 hr (one doubling). DNA was stained with DAPI and tubulin detected by indirect immunofluorescence. Lagging DNA was defined as DAPI-stained material on the anaphase spindle (length >5 μm). p-values from two-sided Fisher's exact test. (**C**) Anaphase cells with displaced spindle. Cells were cultured at 36°C, fixed and stained with Calcofluor to visualize septa. Mean and SD from three independent experiments (*n* > 70 cells per strain and per experiment). p-value from two-sided Fisher's exact test. (**D**) Inter-kinetochore distance. Cells were treated as in *Figure 2G* (*n* = 51 for each mutant). Data from *Figure 2G* are shown for comparison (dotted lines). (**E**) Rad21 ChIP. Exponentially growing cells at 25°C were shifted at 36°C for 3.5 hr (one doubling). Mean and SD from 6 ratios (3 technical replicates from 2 independent experiments). **p ≤ 0.01, *p ≤ 0.05, by two-tailed, one sample *t*-test with 95% confidence interval.

The online version of this article includes the following figure supplement(s) for figure 6:

**Figure supplement 1.** Related to *Figure 6E*.

**Figure supplement 2.** Mimicking the non phosphorylated state of Psm1-S1022 and Mis4-S183 suppressed *pph3Δ psm3*[K105-K106N] phenotypes.

result in an additive effect, consistent with rapamycin and *mip1-R401G* acting in the same pathway. In a similar manner, *psm1-S1022A* showed minimal responsiveness to rapamycin, while *psm1-S1022D* exhibited a discernible response. The involvement of Mis4-S183 phosphorylation was less clear. The *mis4-S183A* mutant did show a reduced response to rapamycin (*Figure 7—figure supplement 1*). However, the phosphomimetic allele *mis4-S183E* did not restore the response to rapamycin, and the double mutants did not reveal a clear genetic interaction, which complicated the interpretation.

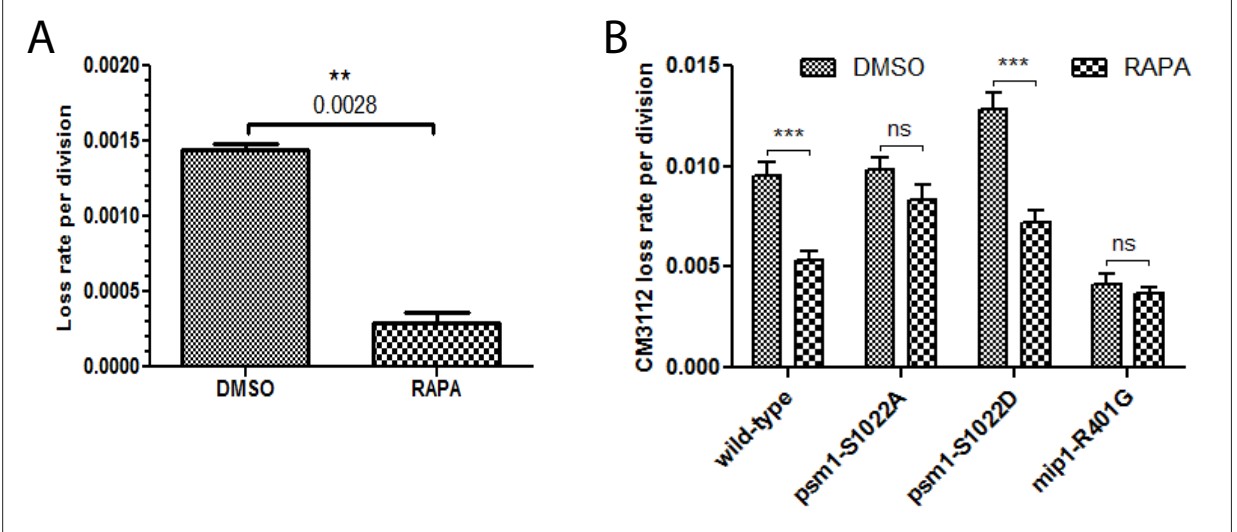

**Figure 7.** Rapamycin reduces the frequency of chromosome loss. (**A**) The rate of chromosome loss was estimated by measuring the breakdown of diploids after growth in medium containing rapamycin (RAPA, 200 ng/ml) or solvent alone (DMSO). Data are mean ± SD, $n = 2$ per group, **p < 0.01, unpaired $t$-test. (**B**) Loss rate of the CM3112 minichromosome. The loss rate was determined in the presence of rapamycin (RAPA, 200 ng/ml) or solvent alone (DMSO). Data are mean ± SEM, $n = 4$ per group ($n = 10$ for the wild-type), ***p < 0.001, repeated measures two-way ANOVA followed by Bonferroni post hoc test.

The online version of this article includes the following figure supplement(s) for figure 7:

**Figure supplement 1.** CM3112 loss rate was determined in the presence of rapamycin (RAPA, 200 ng/ml) or solvent alone (DMSO).

Taken together, these data indicate that rapamycin increases the stability of chromosomes through a pathway involving Mip1 and the phosphorylation of the cohesin subunit Psm1.

## Mis4 and TORC1 control a common set of genes involved in the response to environmental changes

In a previous study, gene expression profiling of *mis4-G1487D* identified a small number of upregulated and downregulated genes within subtelomeric domains (*Dheur et al., 2011*). The genetic link between TORC1 and cohesin prompted us to ask how the two pathways may interact in terms of gene expression. As TORC1 adapts cells to changes in their environment, we looked at the transcriptome by RNA-sequencing of wild-type, *mis4-G1487D*, *mip1-R401G* and the double mutant in various experimental conditions (*Figure 8A*). These included cycling cells in the rich complete medium YES at 25°C (V25) and upon a temperature shift to 36.5°C for one doubling (V36.5) or arrested in G1 (*cdc10-129*) at this temperature (G1_36.5). We also looked upon nitrogen starvation by shifting actively cycling cells from the synthetic medium EMM2 to EMM2 deprived of a nitrogen source (EMM2-N). In this situation, TORC1 activity is restrained, cells arrest in G1 and either mate to enter the reproductive cycle (meiosis) or enter a G0, quiescent state. As no mating partner was available in our experiments, cells homogeneously arrested in G1 after nitrogen deprivation (*Figure 8—figure supplement 1*). RNA-sequencing was performed 24 hr after the shift to EMM2-N (T0-N). One half of the remaining culture was shifted to 36.5°C for 4 days (T4D36.5) while the other half was kept at 25°C for 4 days (T4D25). The analysis of the results is shown in *Figure 8*, and the complete dataset is given in *Supplementary file 2*.

As in the previous study (*Dheur et al., 2011*), a small number of genes were found to be differentially expressed in the *mis4-G1487D* mutant compared to the wild-type (*Figure 8A*). In actively growing cells in rich medium 12 genes were affected, 50 after one cell doubling at 36.5°C and 27 in G1-arrested cells at 36.5°C. The number of differentially expressed genes increased upon nitrogen deprivation (T0-N, 118 genes, T4D25, 125 genes). A similar number of genes were affected in *mis4-G1487D* when nitrogen depleted cells were shifted to 36.5°C for 4 days (T4D36.5, 76 genes). A much larger number of genes were differentially expressed in *mip1-R401G* versus wild-type (*Figure 8A*). As

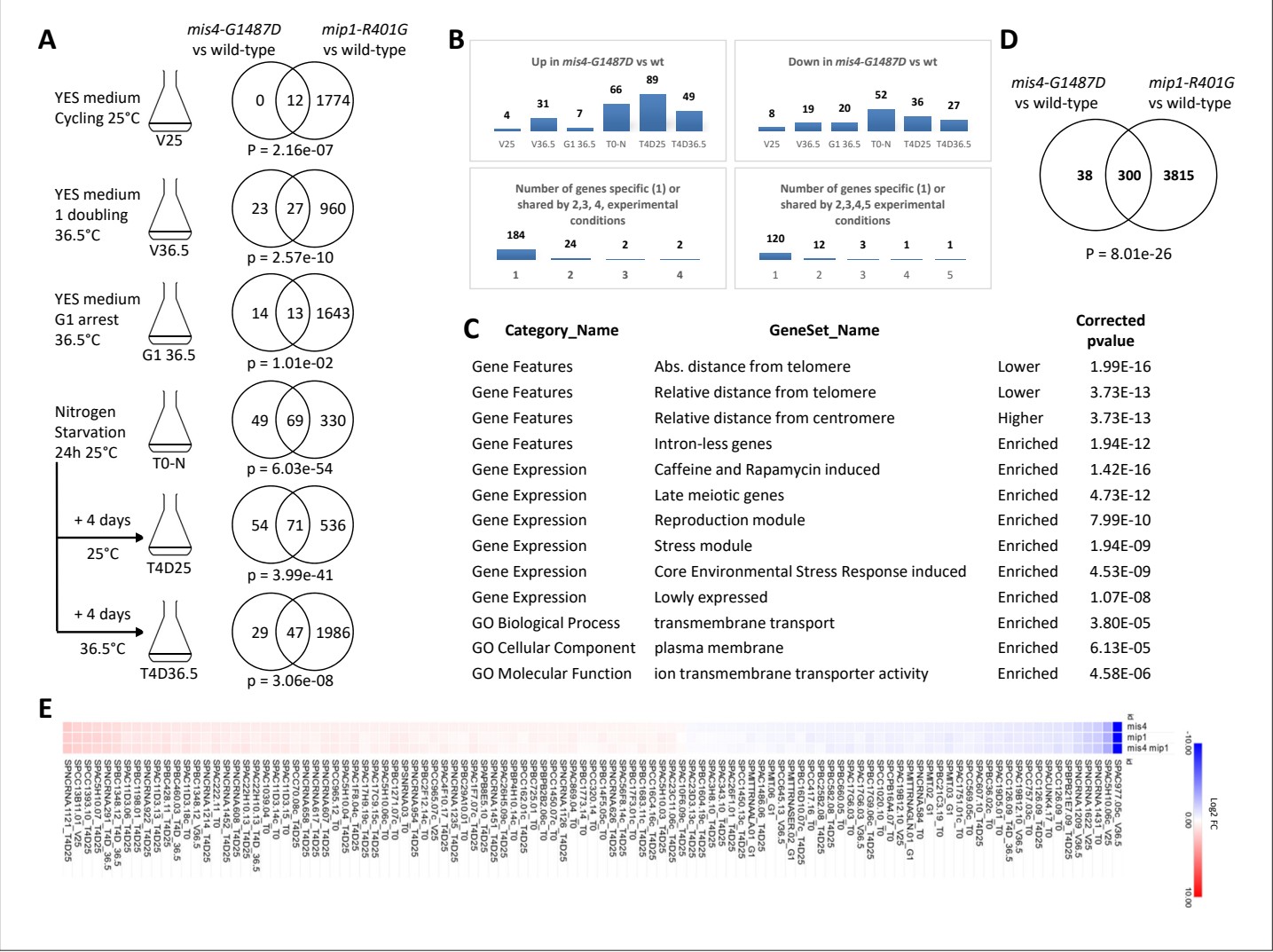

**Figure 8.** *mis4-G1487D* affects gene response to environmental changes. (**A**) RNA-sequencing (biological triplicates) was performed on cells growing in rich medium at 25°C (V25), after one cell doubling at 36.5°C (V36.5) and on G1-arrested cells (*cdc10-129*) at 36.5°C (G1_36.5). For nitrogen starvation, cells grown in EMM2 were deprived from nitrogen for 24 hr at 25°C (T0-N). One half of the culture was shifted to 36.5°C for 4 days (T4D36.5) while the other half was left at 25°C for 4 days (T4D25). The Venn diagrams show the overlap number of differentially expressed genes (FDR <0.05) for each experimental condition. p-value for under- or over-enrichment were calculated based on the cumulative distribution function of the hypergeometric distribution. (**B**) Differentially expressed genes in *mis4-G1487D* across all experiments. There were 408 hits involving 338 genes. The lower graphs show that most of the affected genes are specific to a single experimental condition. (**C**) Gene features and GO-term analysis for the 338 *mis4*-regulated genes. (**D**) Overlap between genes differentially expressed in *mis4-G1487D* and *mip1-R401G* versus wild-type in all experimental conditions. (**E**) Among the 338 genes differentially expressed in *mis4*, 100 show a similar (±10%) fold change in the *mis4*, *mip1*, and *mis4 mip1* mutant strains. See also *Figure 8—figure supplement 6*.

The online version of this article includes the following figure supplement(s) for figure 8:

**Figure supplement 1.** Response to nitrogen starvation and refeeding.

**Figure supplement 2.** Mis4-regulated genes are enriched for tor2 and G0-regulated genes and show a distribution bias towards the ends of chromosomes.

**Figure supplement 3.** Cell survival during quiescence.

**Figure supplement 4.** Quiescence exit at 25°C.

**Figure supplement 5.** Quiescence exit at 36.5°C.

**Figure supplement 6.** Among the 338 genes differentially expressed in *mis4-G1487D*, 100 (107 occurrences, 100 genes) show a similar fold change (±10%) in *mis4-G1487D*, *mip1-R401G*, and the double mutant.

**Figure supplement 7.** Genes differentially expressed in *mis4-G1487D* showing a change in the opposite direction in the *mip1* mutant.

illustrated by the Venn diagrams, a significant proportion of *mis4*-affected genes were also affected in the *mip1* mutant.

Most *mis4*-affected genes were specific to a particular experimental condition (*Figure 8B*). For example, of the genes that are upregulated in the *mis4-G1487D* mutant compared to the wild-type, 184 are specific to a single experimental condition, 24 are upregulated in two experiments, two are upregulated in three experiments, and two are upregulated in four experiments. A similar trend is seen for downregulated genes. Thus, most of the Mis4-sensitive genes are only sensitive under specific culture conditions. These observations suggest that *mis4-G1487D* affects the transcriptional response to environment changes.

In aggregate, 338 genes were differentially expressed in *mis4-G1487D* relative to wild-type across all experiments (*Figure 8B*). The analysis of gene lists with AnGeLi (*Bitton et al., 2015*) indicated that *mis4*-regulated genes showed a localization bias towards chromosome ends (*Figure 8C*, *Figure 8—figure supplement 2*). The response to nitrogen starvation and differentiation (meiosis), as well as enrichment for 'stress genes', were observed, consistent with a defective adaptive response. Genes encoding membrane transporters were significantly enriched, suggesting a role in cell homeostasis. A strong overlap was apparent with a published set of genes whose expression was upregulated in a *tor2* conditional mutant at the restrictive temperature (*Figure 8—figure supplement 2*, *Wei et al., 2021*). A significant overlap was also seen with genes upregulated in G0 (*Zahedi et al., 2023*), suggesting Mis4 may participate in the proper establishment and/or maintenance of the quiescent state (*Figure 8—figure supplement 2*). Cell survival remains high in G0 in the *mis4* and *mip1* mutants (*Figure 8—figure supplements 3 and 4*) but upon refeeding a delay in G0 exit was noticeable (*Figure 8—figure supplement 5*). This phenotype was quite strong for *mip1-R401G* and more subtle for the *mis4* mutant.

Most (300/338) genes affected by *mis4-G1487D* were also affected by *mip1-R401G* (*Figure 8D*). If we go into more detail, we can identify a large group of genes (100/300) for which the quantitative variations were similar, moving in the same direction, with little or no additive effect in the double mutant (*Figure 8E*, *Figure 8—figure supplement 6*). For this particular set of genes the epistatic relationship suggests that Mis4 and TORC1 may act in the same pathway. In other instances, the changes in gene expression relative to the wild-type were in opposite directions in *mis4* and *mip1* mutants (64 occurrences, 62 genes, *Figure 8—figure supplement 7*). Overall, these data show that Mis4 and TORC1 functionally interact for the regulation of genes involved in the response to environmental changes.

## Discussion

The data presented in this work provide evidence for a regulation of cohesin by the TOR signalling pathway. This is, to the best of our knowledge, the first report to demonstrate such a relationship. Previously, a link had been established between the two pathways, but in the opposite direction. mTORC1 is indeed downregulated in cells from patients with Roberts syndrome, caused by mutational alteration of the ESCO2 cohesin acetyl-transferase (*Xu et al., 2013*). The downregulation of the cohesin loader NIPBL in breast cancer cells induced cell cycle arrest, apoptosis and autophagy through the caspase 3 and mTOR signalling pathways (*Zhou et al., 2017*). The first hint for a regulation of cohesin by the TOR pathway came from the observation in fission yeast that the growth of a conditional mutant of *mis4* was enhanced by rapamycin (*Sajiki et al., 2018*), although the underlying molecular mechanisms remained unclear. In mammalian species, a fraction of mTOR controls nuclear processes, notably transcription (*Zhao et al., 2024*). Likewise, fission yeast TORC1 is involved in the regulation of facultative heterochromatin (*Wei et al., 2021*; *Hirai et al., 2023*) and transcriptional processes (*Laribee and Weisman, 2020*). The present work establishes, for the first time, a close link between cohesin and the TOR pathway, suggesting the exciting possibility that extracellular signals could remodel the functional architecture of chromosomes during differentiation or adaptation processes.

### Mis4 and Psm1 as downstream targets of TORC1

We present evidence showing that cohesin and TORC1 components co-purify from protein extracts and Mis4-S183 and Psm1-S1022 are less phosphorylated in *mip1* mutants. Non-phosphorylatable

mutants recapitulated most of the effects of TORC1 downregulation while mutants mimicking the phosphorylated state produced phenotypes similar to TORC1 hyper activation. Whether cohesin and Mis4 are substrates of TORC1 or other kinases regulated by TORC1 is currently unknown. Meanwhile, our work provides the first evidence that cohesin is a downstream effector of TORC1. It is worth mentioning that TORC1 components were found to co-purify with the budding yeast cohesin loader (*Mattingly et al., 2022*), suggesting that the link between cohesin and TORC1 may extend to other species.

Both phosphorylation events on Mis4 and Psm1 have been reported in the literature but their biological functions are unknown (*Kettenbach et al., 2015*; *Swaffer et al., 2018*; *Tay et al., 2019*; *Halova et al., 2021*). Mis4-S183 is located within an unstructured, flexible region connecting the Ssl3 interaction domain and the hook domain that provides Mis4 catalytic activity. It was proposed that phosphorylation at CDK sites within the unstructured linker may affect the flexibility of the complex and modulate its activity (*Chao et al., 2015*). Mis4-S183 and the CDK consensus are conserved in hNIPBL and the residue was found phosphorylated in a number of phosphoproteomic studies (*Ruse et al., 2008*; *Dephoure et al., 2008*; *Pan et al., 2009*; *Mayya et al., 2009*; *Christensen et al., 2010*; *Rigbolt et al., 2011*; *Beli et al., 2012*). Cryo-EM structures of cohesin bound to its loader and DNA revealed that the SMC coiled coils are folded around their elbow (*Collier et al., 2020*; *Shi et al., 2020*). The S1022 residue of Psm1 lies close to the 'joint' region below the ATPase heads that the 'hinge' domain contacts when cohesin is in its folded conformation. Whether the phosphorylation of Psm1-S1022 affects the outcome of DNA transactions remain to be investigated.

## TORC1 activity restrains cohesin binding to CARs

Reducing TORC1 activity by the *mip1-R401G* mutation enhanced cohesin binding to CARs. Remarkably, a similar effect was observed within a single cell cycle of rapamycin treatment. Increased cohesin binding at CARs occurred in two genetic contexts of sensitized cohesin loading and with a wild-type cohesin loading machinery. The most straightforward interpretation is that TORC1 restrains cohesin loading onto chromosomes. One distinct possibility would be that TORC1 downregulation promotes cohesin accumulation at CARs without de novo loading. It was proposed that CARs correspond to the borders of yeast TAD-like structures (*Costantino et al., 2020*). Increased abundance of cohesin at CARs may result from the accumulation of cohesin with loop extrusion activity at TAD boundaries. Considering that loop extrusion requires the NIPBL–cohesin holocomplex (*Davidson et al., 2019*), it is interesting to note that Mis4 binding to CARs was similarly enhanced. TORC1 may also impinge on cohesin sliding by transcription-related processes (*Lengronne et al., 2004*; *Schmidt et al., 2009*) and/or influence cohesin accumulation through the phosphorylation of chromatin-bound proteins. Future work will aim at clarifying these issues.

## TORC1 and chromosome segregation

Cells with an impaired cohesin loading machinery experience chromosome segregation defects during mitosis. We have shown here that the intensity of the defect is dependent on TORC1 activity. The most prominent defect is the merotelic attachment of kinetochores which, when left uncorrected, lead to aberrant chromosome segregation. Although the downregulation of TORC1 reduced the frequency of aberrant anaphases, sister-chromatid cohesion remained largely impaired, even though cohesin binding to CARs in the *mis4* mutant was restored to near wild-type levels in cells arrested at the G1/S boundary. The increase in chromosome bound cohesin may have enhanced cohesion establishment during S-phase. Indeed the inter-kinetochore distance was reduced but sister-chromatid cohesion remained overall defective at a distal, chromosome arm locus. Consistently, cohesin acetylation was not improved by TORC1 downregulation. The acetylation of Psm3, mediated by Eso1 in fission yeast, counteracts the cohesin releasing activity of Wapl and is thought to stabilize cohesin binding and cohesion (*Feytout et al., 2011*; *Kagami et al., 2011*; *Vaur et al., 2012*; *Birot et al., 2017*). However, deletion of the *wpl1* gene does not rescue the *mis4-G1487D* mutant and *S. pombe* cells can survive with non-detectable acetylation or in a context in which Psm3 cannot be acetylated (*Feytout et al., 2011*; *Birot et al., 2020*). The primary defect of *mis4-G1487D* may therefore stem from inability to generate sister-chromatid cohesion rather than its maintenance. In vitro assays have suggested that sister-chromatid capture by cohesin is a two-step process catalysed by Mis4 (*Murayama et al., 2018*). The downregulation of TORC1 may increase the residual loading activity of Mis4 but second DNA

capture may remain largely deficient, resulting in poor sister-chromatid cohesion. Increased cohesin binding at peri-centromeres may improve chromosome segregation through other routes. Cohesin organizes the peri-centromere regions in budding yeast (*Lawrimore et al., 2018*; *Paldi et al., 2020*). Local depletion of cohesin at centromeres in fission yeast lead to a high incidence of merotelic attachments (*Bernard et al., 2001*). A recent study revealed that vertebrate kinetochores are bipartite structures and highlighted a role for cohesin in bridging to the two parts into a functional unit (*Sacristan et al., 2024*). Lastly, enhanced cohesin binding to centromeres may facilitate the correction of merotelic attachments by the Aurora B kinase. The localization of Aurora B relies on histone H3–T3 phosphorylation by Haspin whose localization to centromeres depends largely on the cohesin subunit Pds5 in fission yeast (*Yamagishi et al., 2010*; *Goto et al., 2017*).

That chromosome segregation defects increased with TORC1 activity suggests the intriguing possibility that the fidelity of chromosome segregation could be modulated by environmental cues. Chromosome segregation defects in the *mis4* mutant were most intense in the complete rich medium YES than in the synthetic medium EMM2. The use of glutamate, a nitrogen source slightly poorer than ammonia (*Davie et al., 2015*) further reduced chromosome segregation defects. Furthermore, Rapamycin reduced the frequency of chromosome loss in wild-type cells in a pathway involving the phosphorylation state of Psm1-S1022. It is striking that rapamycin increased the stability of chromosomes during vegetative growth. The idea that nutritional inputs affect the stability of chromosomes could make sense in ecological terms. Microbial species in the wild compete with each other. When nutrients are abundant, it may be preferable to divide in order to colonize the niche rapidly, even at the expense of a few cells. Conversely, when nutrients become scarce, survival of the species may rely more heavily on the fitness of individual cells.

TORC1-based regulation of cohesin may be relevant to preparing cells for meiosis. Since nitrogen deprivation stimulates meiosis initiation, subsequent TORC1 down-regulation may regulate the cohesin complex, preparing the chromosomes for fusion and meiosis. A recent phosphoproteomic study conducted by Sophie Martin's laboratory showed that Mis4-S107 phosphorylation increases during cellular fusion (*Bérard et al., 2024*). It is unknown whether the phosphorylation of S107 is controlled by TORC1 signalling. As the phosphorylation of Mis4-S183 and Psm1-S1022 was not detected in these experiments, the potential involvement of the TORC1-cohesin axis in the sexual programme remains to be investigated.

## Mis4 and TORC1 control the gene response to environmental changes

In a previous study we reported that *mis4-G1487D* affects the expression of genes located in subtelomeric domains (*Dheur et al., 2011*). Here, we extend the study by challenging cells with various culture conditions. Several conclusions can be drawn. Very few genes were affected when cells were actively cycling in rich media, when TORC1 activity is high. The gene list was much longer upon nitrogen deprivation, when TORC1 activity is repressed, and the list of genes evolved with time spent in G0. In aggregate 338 genes were affected in *mis4-G1487D* when compared with wild-type with a bias towards chromosome ends and far from centromeres. Lastly, most genes misregulated in *mis4-G1487D* were misregulated in the *mip1-R401G* mutant suggesting Mis4 and Mip1 act in a same pathway. Paradoxically, the *mis4* and *mip1* mutants remained viable during G0 although a delay was noticeable upon re-entry into the cell cycle at elevated temperature. A defect in quiescence exit was previously reported when the *mis4* gene is affected and the phenotype appears strongly dependent on the allele considered (*Sajiki et al., 2009*; *Suma et al., 2024*). It must be kept in mind that *mis4* mutants were originally screened for a defect during vegetative growth. It would be interesting to screen specifically for *mis4* alleles affecting cell survival in quiescence.

Given the role of cohesin and its loader in chromosome folding, it is tempting to speculate that TORC1 might direct cohesin to establish a chromosome architecture conducive to transcriptional responses to environmental changes. In yeast, cohesin distribution alone predicts chromatin organization (*Yuan et al., 2024*). If increased binding of Mis4 and cohesin at CARs does reflect extruding cohesin complexes that have reached TAD boundaries, chromosomes might be more compacted when TORC1 is downregulated. Conversely, high TORC1 would promote the opposite. Even if this hypothesis turned out to be correct, it does not explain why regulated genes show a bias towards chromosome ends. Chromosomes in fission yeast adopt a Rabl configuration in interphase (*Mizuguchi et al., 2015*). Centromeres cluster at the spindle pole bodies (SPBs) while telomeres cluster in a few

patches at the opposing hemisphere near the nuclear periphery. These constraints generate specific chromosomal regions with distinct molecular environments. Chromosome conformation capture and the mapping of DNA-binding sites of inner nuclear membrane proteins have provided evidence of functional sub-nuclear environments that correlate with gene expression activity (*Steglich et al., 2012*; *Grand et al., 2014*). The studies suggest more internal locations of actively expressed genes, and nuclear peripheral localization of poorly expressed genes. Chromosome regions containing gene clusters that are upregulated upon nitrogen starvation re-position from the nuclear periphery to the interior (*Alfredsson-Timmins et al., 2009*). In this regard, Mis4-regulated genes tend to be lowly expressed and enriched for nitrogen responsive genes. Modifying chromosome compaction may alter the Rabl configuration or generate forces that modify the properties of the inner nuclear envelope, leading to misregulated gene expression. Indeed, loss of cohesin causes an increase in inter-chromosomal contacts and a reduction in chromosomal territoriality (*Mizuguchi et al., 2014*).

Cohesin and TORC1 have been implicated with a wide range of human pathologies, including cancer (*Yoon, 2020*; *Waldman, 2020*). Rapamycin has attracted much attention for its putative ability to mimic caloric restriction, resulting in improved health and lifespan (*Bjedov and Partridge, 2011*). Our study suggests the intriguing possibility that some effects are mediated by the ability of cohesin to remodel chromosomes in response to various extracellular signals, including nutrients.

# Materials and methods

## Key resources table

| Reagent type (species) or resource | Designation | Source or reference | Identifiers | Additional information |
|---|---|---|---|---|
| Antibody | Rabbit polyclonal against *S. pombe* Rad21 | PMID:28438891 | | Western blot, 0.8 µg/ml Immunoprecipitation, 2 µg |
| Antibody | Rabbit polyclonal against *S. pombe* Rad21-T262p | PMID:31895039 | | Western blot, 2.7 µg/ml |
| Antibody | Rabbit polyclonal against Psm1 | PMID:21189291 | | Western blot 1:1000 |
| Antibody | Rabbit polyclonal against Psm3 | PMID:21300781 | | Western blot, 0.4 µg/ml |
| Antibody | Rabbit polyclonal against Psm3-K106Ac | PMID:21300781 | | Western blot, 1 µg/ml |
| Antibody | Rabbit polyclonal against Psm1-S1022p | This paper, MATERIALS AND METHODS section | | Western blot, 1.12 µg/ml |
| Antibody | Rabbit polyclonal against Mis4-S183p | This paper, Materials and methods section | | Western blot, 0.55 µg/ml |
| Antibody | Mouse monoclonal anti-tubulin antibody TAT1 | PMID:2606940 | | Immunofluorescence, 1:50 |
| Antibody | Rabbit polyclonal anti-GFP | | Molecular Probes Cat# A-11122, RRID:AB_221569 | Western blot, 2 µg/ml Immunoprecipitation, 2 µg Immunofluorescence, 5 µg/ml |
| Antibody | Mouse monoclonal anti-GFP | | Roche Cat# 11814460001, RRID:AB_390913 | Western blot, 0.4 µg/ml Immunoprecipitation, 2 µg |
| Antibody | anti-PK (monoclonal mouse anti V5 tag) | | Bio-Rad Cat# MCA1360, RRID:AB_322378 | Western blot, 1 µg/ml Immunoprecipitation, 2 µg |
| Antibody | Mouse monoclonal anti-FLAG | | (Sigma-Aldrich Cat# F1804, RRID:AB_262044) | Western blot, 1.7 µg/ml Immunoprecipitation, 2 µg |
| Antibody | Mouse monoclonal anti-myc | | Santa Cruz Biotechnology Cat# sc-40, RRID:AB_627268 | Western blot, 0.2 µg/ml |
| Antibody | Rabbit anti-phospho-p70 S6K antibody | | Cell Signaling Technology Cat# 9206, RRID:AB_2285392 | Western blot, 0.7 µg/ml |

## Strains, media, and genetic techniques

General fission yeast methods, reagents and media are described in *Moreno et al., 1991*. All strains are listed in *Supplementary file 1*. Experiments were carried out using YES medium unless otherwise stated. For cell growth assays serial dilutions of cells were spotted on YES medium and the plates incubated at the indicated temperatures for 3 days unless otherwise stated. EMM-GLU is EMM2 in which $NH_4Cl$ was replaced with 20 mM glutamate. Rapamycin (Calbiochem, 4.57 mg/ml stock in DMSO) was added to the culture medium to 200 ng/ml. For nitrogen deprivation, cycling cells in EMM2 (~5–$10^6$–$10^7$ cells/ml) were washed three times in EMM2-N and incubated in EMM2-N at the same density. Cell cycle arrest in YES medium was achieved by shifting cycling *cdc10-129* cells (~$10^7$ cells/ml) at 36°C for 3.5 hr. Hydroxyurea (HU) was used to arrest cells in early S-phase. HU (12 mM) was added to cycling cells in YES (~$10^7$ /ml) and shifted to 36.5°C for 3.5 hr. All cell cycle arrests were checked by flow cytometry analysis of DNA content. The genetic screen for *mis4* suppressors was described in *Birot et al., 2020*. The suppressors fell into four linkage groups. To identify the mutated loci, genomic DNA was extracted from one mutant from each group and from the wild-type *S. pombe* reference strain and co-hybridized to tiling arrays as described (*Birot et al., 2020*). For the *mip1* group (five alleles) the mutated site in each strain was identified by PCR followed by DNA sequencing. Phospho-mutants were generated using the SpEDIT method (*Torres-Garcia et al., 2020*). The introduction of the desired changes was confirmed by PCR followed by DNA sequencing. Gene mapping, sequence and annotations were from Pombase (*Rutherford et al., 2024*. Chromosome loss assays were as described (*Allshire et al., 1995*) using the calculation method of *Kipling and Kearsey, 1990*). Cells containing the CM3112 minichromosome are Ade+ and form white colonies on YES medium. Loss of the minichromosome generates Ade- cells that form red colonies. White colonies grown on YES medium (5–10) were picked, resuspended in 1 ml of YES+Adenine medium, and the cell density was measured. Cells were diluted to $10^4$/ml in YES+Adenine and incubated at 25°C for 3 days. Cells (500–1000) were plated on YES plates at days 0 and 3 to estimate the proportion of cells containing the minichromosome. The rate of chromosome loss per division was calculated using the formula: loss rate = $1 - (F/I)^{1/N}$, where $F$ is the final percentage of cells bearing the minichromosome, $I$ is the initial percentage of cells carrying the minichromosome, and $N$ is the number of generations between $I$ and $F$. The same method was applied to measure the breakdown of diploid strains but with two successive rounds of growth in liquid YES+Adenine medium before the final plating.

## Cytological techniques

DNA content was measured by flow cytometry with an Accuri C6 Flow cytometer after Sytox Green staining of ethanol-fixed cells (*Knutsen et al., 2011*). Data were presented using the FlowJo software. Indirect immunofluorescence was done as described (*Birot et al., 2020*). Cells were imaged using a Leica DMRXA microscope and a 63X objective. Measurements were made using MetaMorph software.

Live cell analysis was performed in an imaging chamber (CoverWell PCI-2.5, Grace Bio-Labs, Bend, OR) filled with 1 ml of 1% agarose in minimal medium and sealed with a 22 × 22 mm glass coverslip. Time-lapse images of Z stacks (maximum five stacks of 0.5 µm steps, to avoid photobleaching) were taken at 60-s intervals. Images were acquired with a CCD Retiga R6 camera (QImaging) fitted to a DM6B upright microscope with a x63 objective, using MetaMorph as a software. Intensity adjustments were made using the MetaMorph, Image J, and Adobe Photoshop packages (Adobe Systems France, Paris, France). To determine the percentage of mispositioned spindles, cells were fixed in 3.7% formaldehyde for 6 min at room temperature, washed twice in PBS, and observed in the presence of DAPI/calcofluor.

The position of the SPBs, kinetochores and centromeres were determined by visualization of the Cdc11–GFP, Ndc80–GFP and Mis6-RFP signals. Maximum intensity projections were prepared for each time point, with the images from each channel being combined into a single RGB image. These images were cropped around the cell of interest, and optional contrast enhancement was performed in MetaMorph, Image J or Photoshop where necessary. The cropped images were exported as 8-bit RGB-stacked TIFF files, with each frame corresponding to one image of the time-lapse series. For all channels, custom peak detection was performed. The successive positions of the SPBs and kinetochores/centromeres were determined.

For the measurement of inter-kinetochore distance, cells were grown overnight at 25°C, then shifted to 36.5°C for 5 hr and subsequently chilled at 4°C for 30 min. At this stage, as previously

shown (*Gachet et al., 2008*), metaphase spindles collapsed, microtubules were absent, and sister kinetochores were frequently misaligned relative to the two SPBs. After the cold shock, cells were fixed in 3.7% formaldehyde for 6 min at room temperature, washed twice in PBS, stained with DAPI and the distance between Mis6-RFP signals was measured.

## Antibodies, protein extracts, immunoprecipitation, western blotting, phosphatase treatment, and ChIP

Rabbit polyclonal antibodies against Rad21, Psm1, Psm3, Psm3-K106Ac, and Rad21-T262 have been described previously (*Feytout et al., 2011*; *Dheur et al., 2011*; *Birot et al., 2020*; *Birot et al., 2017*). The mouse monoclonal anti-tubulin antibody TAT1 is from *Woods et al., 1989*. Anti-Psm1-S1022p and anti-Mis4-S183p antibodies were raised by Biotem (Apprieu, France). Rabbits were immunized with the KLH-coupled peptides. Sera were immune-depleted by affinity with the non-phosphorylated form of the peptide and antibodies were affinity purified against the phosphorylated peptide. Other antibodies were of commercial source. Rabbit polyclonal anti-GFP A11122 (Molecular Probes), mouse monoclonal anti-GFP (Roche), anti-PK antibodies (monoclonal mouse anti V5 tag, AbD serotec), mouse monoclonal anti-FLAG (Sigma), and mouse monoclonal anti-myc (Santa-Cruz). For the detection of Psk1 (Thr-415) phosphorylation, protein extracts were prepared by the TCA method and western blots probed with the anti-phospho-p70 S6K antibody (cat. no. 9206, Cell Signaling Technology) as described (*Morozumi et al., 2021*). Protein extracts, immunoprecipitation, cell fractionation and western blotting were as described (*Birot et al., 2020*). For quantitative western blot analyses, signals were captured with the ChemiDoc MP Imaging System and quantified using the Image Lab software. On beads phosphatase treatment of Mis4-GFP was done as described (*Birot et al., 2017*). ChIP was as described (*Birot et al., 2020*) using anti-FLAG, anti-GFP (A11122), and anti-Rad21 antibodies.

## RNA sequencing

Total RNA from biological triplicate samples was subjected to quality control with Agilent Tapestation according to the manufacturer's instructions. To construct libraries suitable for Illumina sequencing, rRNA was depleted using the Ribominus kit (Thermo Fisher) starting with 2000 ng total RNA and then followed by the Illumina stranded ligation sample preparation protocol starting with 100 ng rRNA depleted RNA. The protocol includes fragmentation, denaturation of RNA, cDNA synthesis, ligation of adapters, and amplification of indexed libraries. The yield and quality of the amplified libraries were analysed using Qubit (Thermo Fisher) and the quality of the libraries was checked by using the Agilent Tapestation. The indexed cDNA libraries were normalized and combined, and the pools were sequenced on the Illumina Nextseq 2000 machine using a P3 100 cycle sequencing run, producing a sequencing length of 58 base paired end reads with dual index.

## RNA-Seq data processing and analysis

Bcl files were converted and demultiplexed to fastq using the bcl2fastq v2.20.0.422 program. STAR 2.7.10a (*Dobin et al., 2013*) was used to index the *S. pombe* reference genome (ASM294v2) and align the resulting fastq files. Mapped reads were then counted in annotated exons using feature-Counts v1.5.1 (*Liao et al., 2014*). The gene annotations (Schizosaccharomyces_pombe.ASM294v2.35. gff3) and reference genome were obtained from Ensembl Fungi. The count table from featureCounts was imported into R/Bioconductor and differential gene expression was performed using the EdgeR (*Robinson et al., 2010*) package and its general linear models pipeline. For the gene expression analysis genes with no or very low expression were filtered out using the filterByExpr function and subsequently normalized using TMM normalization. Genes with an FDR adjusted p-value <0.05 were termed significantly regulated. Venn diagrams were generated using jvenn (*Bardou et al., 2014*). The p-values for under- or over-enrichment were calculated based on the cumulative distribution function of the hypergeometric distribution. The total number of genes was set to 6403, which corresponds to the total number of genes that were detected across all our experiments.

## Mass spectrometry

Sample preparation and protein digestion. Protein samples were solubilized in Laemmli buffer and proteins were deposited onto SDS–PAGE gel. After colloidal blue staining, each lane was cut out from the gel and was subsequently cut in 1 mm × 1 mm gel pieces. Gel pieces were destained in 25 mM

ammonium bicarbonate 50% ACN, rinsed twice in ultrapure water and shrunk in ACN for 10 min. After ACN removal, gel pieces were dried at room temperature, covered with the trypsin solution (10 ng/µl in 50 mM NH₄HCO₃), rehydrated at 4°C for 10 min, and finally incubated overnight at 37°C. Spots were then incubated for 15 min in 50 mM NH4HCO3 at room temperature with rotary shaking. The supernatant was collected, and an $H_2O$/ACN/HCOOH (47.5:47.5:5) extraction solution was added onto gel slices for 15 min. The extraction step was repeated twice. Supernatants were pooled and dried in a vacuum centrifuge. Digests were finally solubilized in 0.1% HCOOH.

## nLC–MS/MS analysis and label-free quantitative data analysis

Peptide mixture was analysed on a Ultimate 3000 nanoLC system (Dionex, Amsterdam, The Netherlands) coupled to a Electrospray Orbitrap Fusion Lumos Tribrid Mass Spectrometer (Thermo Fisher Scientific, San Jose, CA). Ten microliters of peptide digests were loaded onto a 300-µm-inner diameter × 5-mm C18 PepMapTM trap column (LC Packings) at a flow rate of 10 µl/min. The peptides were eluted from the trap column onto an analytical 75 mm id × 50-cm C18 Pep-Map column (LC Packings) with a 5–27.5% linear gradient of solvent B in 105 min (solvent A was 0.1% formic acid and solvent B was 0.1% formic acid in 80% ACN) followed by a 10 min gradient from 27.5% to 40% solvent B. The separation flow rate was set at 300 nl/min. The mass spectrometer operated in positive ion mode at a 2-kV needle voltage. Data were acquired using Xcalibur software in a data-dependent mode. MS scans ($m/z$ 375–1500) were recorded in the Orbitrap at a resolution of $R$ = 120,000 (@ $m/z$ 200) and an AGC target of $4 \times 10^5$ ions collected within 50 ms. Dynamic exclusion was set to 60 s and top speed fragmentation in HCD mode was performed over a 3-s cycle. MS/MS scans were collected in the Orbitrap with a resolution if 30,000 and maximum fill time of 54 ms. Only +2 to +7 charged ions were selected for fragmentation. Other settings were as follows: no sheath nor auxiliary gas flow, heated capillary temperature, 275°C; normalized HCD collision energy of 30%, isolation width of 1.6 $m/z$, AGC target of $5 \times 10^4$ and normalized AGC target of 100%. Monoisotopic precursor selection was set to Peptide and an intensity threshold was set to $2.5 \times 10^4$.

## Database search and results processing

Data were searched by SEQUEST and BYONIC through Proteome Discoverer 2.5 (Thermo Fisher Scientific Inc) against an *S. pombe* uniprot database (5,098 entries in v2021-01). Spectra from peptides higher than 5000 Da or lower than 350 Da were rejected. Precursor Detector node was included. Search parameters were as follows: mass accuracy of the monoisotopic peptide precursor and peptide fragments was set to 10 ppm and 0.02 Da, respectively. Only b- and y-ions were considered for mass calculation. Sequest HT was used as the search algorithm: Oxidation of methionines (+16 Da), methionine loss (–131 Da), methionine loss with acetylation (–89 Da), protein N-terminal acetylation (+42 Da) and phosphorylation of serine, threonine, and tyrosine (+80 Da) were considered as variable modifications while carbamidomethylation of cysteines (+57 Da) was considered as fixed modification. Two missed trypsin cleavages were allowed. Peptide validation was performed using Percolator algorithm (*Käll et al., 2007*) and only 'high confidence' peptides were retained corresponding to a 1% False Positive Rate at peptide level. Peaks were detected and integrated using the Minora algorithm embedded in Proteome Discoverer. Normalization was performed based on total peptide amount. Protein ratios were calculated as the median of all possible pairwise peptide ratios. A *t*-test was calculated based on background population of peptides or proteins. Quantitative data were considered at peptide level.

## Acknowledgements

We would like to thank Tomoyuki Fukuda, Yuichi Morozumi, Janni Petersen, Kazuhiro Shiozaki, Ronit Weisman, Mitsuhiro Yanagida, and the National BioResource Project for providing *S. pombe* strains and advices. Keith Gull for the gift of anti-tubulin antibodies. This work was supported by the Centre National de la Recherche Scientifique, l'Université de Bordeaux, la Région Aquitaine, la Fondation ARC pour la recherche contre le cancer (ARCPJA2021060003810) and La Ligue contre le Cancer. Dorian Besson was supported by the International Doctorate Program – IdEx University of Bordeaux and a fellowship from the Fondation ARC. Adele Marston is funded through a Wellcome Investigator Award [220780] and the Wellcome Discovery Research Platform for Hidden Cell Biology [226791].

Work in the Karl Ekwall laboratory was financed by the Swedish research council (VR-MH) and the Swedish Cancer Society (CF). We also would like to thank BEA, the core facility for Bioinformatics and Expression Analysis, which is supported by the board of research at the Karolinska Institutet and the research committee at the Karolinska hospital.

## Additional information

### Competing interests

Adèle L Marston: Senior editor, *eLife*. The other authors declare that no competing interests exist.

### Funding

| Funder | Grant reference number | Author |
|---|---|---|
| Fondation ARC pour la Recherche sur le Cancer | ARCPJA2021060003810 | Jean-Paul Javerzat |
| La Ligue Contre le Cancer | | Jean-Paul Javerzat |
| International PhD Programme | IdEx University of Bordeaux | Dorian Besson |
| Fondation ARC pour la Recherche sur le Cancer | | Dorian Besson |
| Wellcome Investigator Award | 10.35802/220780 | Adèle L Marston |
| Wellcome Discovery Research Platform for Hidden Cell Biology | 10.35802/226791 | Adèle L Marston |
| Swedish Research Council | | Karl Ekwall |
| Cancerfonden | | Karl Ekwall |

The funders had no role in study design, data collection, and interpretation, or the decision to submit the work for publication. For the purpose of Open Access, the authors have applied a CC BY public copyright license to any Author Accepted Manuscript version arising from this submission.

### Author contributions

Dorian Besson, Data curation, Formal analysis, Investigation, Writing – review and editing; Sabine Vaur, Sylvie Tournier, Yannick Gachet, Data curation, Formal analysis, Validation, Investigation, Visualization, Methodology, Writing – review and editing; Stéphanie Vazquez, Investigation; Adrien Birot, Formal analysis, Investigation, Writing – review and editing; Stéphane Claverol, Data curation, Formal analysis, Investigation, Methodology, Writing – review and editing; Adèle L Marston, Formal analysis, Funding acquisition, Investigation, Writing – review and editing; Anastasios Damdimopoulos, Data curation, Formal analysis, Validation, Investigation, Methodology, Writing – review and editing; Karl Ekwall, Conceptualization, Resources, Data curation, Formal analysis, Supervision, Funding acquisition, Validation, Investigation, Visualization, Methodology, Writing – review and editing; Jean-Paul Javerzat, Conceptualization, Resources, Data curation, Formal analysis, Supervision, Funding acquisition, Validation, Investigation, Visualization, Methodology, Writing – original draft, Project administration, Writing – review and editing

### Author ORCIDs

Sylvie Tournier ⬤ https://orcid.org/0000-0002-7869-6891
Adrien Birot ⬤ https://orcid.org/0000-0002-8299-8518
Adèle L Marston ⬤ https://orcid.org/0000-0002-3596-9407
Karl Ekwall ⬤ https://orcid.org/0000-0002-3029-4041
Jean-Paul Javerzat ⬤ https://orcid.org/0000-0002-9671-6753

Reviewer #1 (Public review): https://doi.org/10.7554/eLife.108275.3.sa1

Reviewer #2 (Public review): https://doi.org/10.7554/eLife.108275.3.sa2

Author response https://doi.org/10.7554/eLife.108275.3.sa3

## Additional files

### Supplementary files

MDAR checklist

Supplementary file 1. Strain list.

Supplementary file 2. Differentially expressed genes in *mis4-G1487D*, *mip1-R401G* and in the double mutant.

### Data availability

RNA sequence data were deposited in GEO (accession number GSE316645). All data generated or analysed during this study are included in the manuscript and supporting files; uncropped western blots images have been provided in source data files.

The following dataset was generated:

| Author(s) | Year | Dataset title | Dataset URL | Database and Identifier |
|---|---|---|---|---|
| Besson D, Vaur S, Vazquez S, Tournier S, Gachet Y, Birot A, Claverol S, Marston A, Damdimopoulos A, Ekwall K, Javerzat JP | 2026 | Interplay between cohesin and TORC1 links chromosome segregation and gene expression to environmental changes | https://www.ncbi.nlm.nih.gov/geo/query/acc.cgi?acc=GSE316645 | NCBI Gene Expression Omnibus, GSE316645 |

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
