## [Editor Report · eLife Assessment]

This **important** study presents a **compelling** link between nutrient signaling and chromosome regulation, demonstrating that reduced activity in a central nutrient-sensing pathway improves chromosome stability and alters gene expression through effects on cohesin. The **convincing** evidence from a combination of genetic, biochemical and cell biological approaches supports a model in which TORC1-dependent phosphorylation of Mis4 and the cohesin subunit Psm1/Smc1 can modulate cohesin loading to enhance faithful chromosome transmission. While the underlying mechanisms and biological importance of this newly described circuit are not yet fully known, the overall body of evidence is strong and supports the main conclusions.

---

## [Referee Report · Reviewer #1 (Public review)]

Summary:

In this study, Besson et al. investigate how environmental nutrient signals regulate chromosome biology through the TORC1 signaling pathway in *Schizosaccharomyces pombe*. Specifically, the authors explore the impact of TORC1 on cohesin function-a protein complex essential for chromosome segregation and transcriptional regulation. Through a combination of genetic screens, biochemical analysis, phospho-proteomics, and transcriptional profiling, they uncover a functional and physical interaction between TORC1 and cohesin. The data suggest that reduced TORC1 activity enhances cohesin binding to chromosomes and improves chromosome segregation, with implications for stress-responsive gene expression, especially in subtelomeric regions.

Strengths:

This work presents a compelling link between nutrient sensing and chromosome regulation. The major strength of the study lies in its comprehensive and multi-disciplinary approach. The authors integrate genetic suppression screens, live-cell imaging, chromatin immunoprecipitation, co-immunoprecipitation, and mass spectrometry to uncover the functional connection between TORC1 signaling and cohesin. The use of phospho-mutant alleles of cohesin subunits and their loader provides mechanistic insight into the regulatory role of phosphorylation. The addition of transcriptomic analysis further strengthens the biological relevance of the findings and places them in a broader physiological context. Altogether, the dataset convincingly supports the authors' main conclusions and opens up new avenues of investigation.

Points that remain open but are appropriately discussed by the authors:

(1) The authors propose that nutrient status influences cohesin regulation. While this is not directly tested under defined nutrient conditions (e.g., by systematically examining cohesin dynamics or phosphorylation across nutrient states), the rationale is well explained in the text, and the study provides a strong foundation for addressing this question in future work.

(2) The upstream signaling cascade downstream of TORC1 remains to be fully elucidated. In particular, the identity of the relevant kinases (e.g., whether Sck1/Sck2 or other effectors are involved) and whether TORC1 directly phosphorylates Mis4 or Psm1 are not resolved. The authors acknowledge these mechanistic gaps, which represent logical next steps rather than shortcomings of the current study.

---

## [Referee Report · Reviewer #2 (Public review)]

Summary:

In this study the authors follow up on a previous suppressor screen of a temperature-sensitive allele of mis4 (mis4-G1487D), the cohesin loading factor in *S. pombe*, and identify additional suppressor alleles tied to the *S. pombe* TORC1 complex. Their analysis suggests that these suppressor mutations attenuate TORC1 activity while enhanced TORC1 activity is deleterious in this context. Suppression of TORC1 activity also ameliorates chromosome segregation and spindle defects observed in the mis4-G1487D strain, although some more subtle effects are not reconstituted. The authors provide evidence that this genetic suppression is also tied to the reconstitution of cohesin loading. Moreover, disrupting TORC1 also enhances Mis4/cohesin association with chromatin (likely reflecting enhanced loading) in WT cells while rapamycin treatment can enhance the robustness of chromosome transmission. These effects likely arise directly through TORC1 or its downstream effector kinases as TORC1 co-purifies with Mis4 and Rad21; these factors are also phosphorylated in a TORC1-dependent fashion. Disrupting Sck2, a kinase downstream of TORC1, also suppresses the mis4-G1487D allele while simultaneous disruption of Sck1 and Sck2 enhances cohesin association with chromatin, albeit with differing effects on phosphorylation of Mis4 and Psm1/Scm1. Phosphomutants of Mis4 and Psm1 that mimic observed phosphorylation states identified by mass spectrometry that are TORC1-dependent also suppressed phenotypes observed in the mis4-G1487D background. Lastly, the authors provide evidence that the mis4-G1487D background and TORC1 mutant backgrounds display an overlap in the dysregulation of genes that respond to environmental conditions.

Overall, the authors provide compelling evidence from genetics, biochemistry and cell biology to support a previously unknown mechanism by which nutrient sensing regulates cohesin loading with implications for the stress response. The technical approaches are generally sound, well-controlled, and comprehensive.

The specific points that I raised in the first review have been addressed by changes/additions to the manuscript or have been determined to be beyond the scope of the study by the authors.

One major question that remains open is the relationship between local changes in cohesin loading and gene expression through this TORC1 regulatory signaling pathway and the details of the underlying mechanisms.

---

## [Author Response]

The following is the authors’ response to the original reviews.

**Public Reviews:**

**Reviewer #1 (Public review):**
Summary:In this study, Besson et al. investigate how environmental nutrient signals regulate chromosome biology through the TORC1 signaling pathway in *Schizosaccharomyces pombe*. Specifically, the authors explore the impact of TORC1 on cohesin function - a protein complex essential for chromosome segregation and transcriptional regulation. Through a combination of genetic screens, biochemical analysis, phospho-proteomics, and transcriptional profiling, they uncover a functional and physical interaction between TORC1 and cohesin. The data suggest that reduced TORC1 activity enhances cohesin binding to chromosomes and improves chromosome segregation, with implications for stress-responsive gene expression, especially in subtelomeric regions.Strengths:This work presents a compelling link between nutrient sensing and chromosome regulation. The major strength of the study lies in its comprehensive and multi-disciplinary approach. The authors integrate genetic suppression screens, live-cell imaging, chromatin immunoprecipitation, co-immunoprecipitation, and mass spectrometry to uncover the functional connection between TORC1 signaling and cohesin. The use of phospho-mutant alleles of cohesin subunits and their loader provides mechanistic insight into the regulatory role of phosphorylation. The addition of transcriptomic analysis further strengthens the biological relevance of the findings and places them in a broader physiological context. Altogether, the dataset convincingly supports the authors' main conclusions and opens up new avenues of investigation.Weaknesses:While the study is strong overall, a few limitations are worth noting. The consistency of cohesin phosphorylation changes under different TORC1-inhibiting conditions (e.g., genetic mutants vs. rapamycin treatment) is unclear and could benefit from further clarification. The phosphorylation sites identified on cohesin subunits do not match known AGC kinase consensus motifs, raising the possibility that the modifications are indirect. The study relies heavily on one TORC1 mutant allele (mip1-R401G), and additional alleles could strengthen the generality of the findings. Furthermore, while the results suggest that nutrient availability influences cohesin function, this is not directly tested by comparing growth or cohesin dynamics under defined nutrient conditions.

We thank the reviewer for his overall positive assessment and constructive criticism. We broadly agree with the few limitations he pointed out, which we will comment on below.

(1) The consistency of cohesin phosphorylation changes under different TORC1-inhibiting conditions (e.g., genetic mutants vs. rapamycin treatment) is unclear and could benefit from further clarification.

The basis of our study was to search for suppressor mutants, a situation in which an unviable strain becomes viable. It turns out that the suppressor mutants affect TORC1, necessarily in a partial manner given that TORC1 kinase activity is essential for proliferation. Likewise rapamycin partially inhibits TORC1 and does not prevent proliferation of wild-type *S. pombe* cells. TORC1 mutants cause a constitutive decrease in activity with possible adaptive effects, whereas rapamycin is applied for a single cell cycle. In addition, it is known that bona fide TORC1 substrates respond differently to rapamycin. Some phosphosites show acute sensitivity, while others are less sensitive or even insensitive (Kang et al., 2013, PMID: 23888043). Therefore, both hypomorphic TORC1 genetic mutants and rapamycin treatment result in partial inhibition of TORC1 kinase activity. While the lists of affected TORC1 substrates may overlap, they are unlikely to be identical. Furthermore, the phosphorylation level of the relevant substrates is not necessarily altered to the same extent. Nevertheless, both conditions suppress the heatsensitive phenotype of the mis4 mutant, although the suppressor effect of rapamycin is weaker. Consequently, some phosphorylation sites involved in mis4-ts suppression may behave similarly in rapamycin and TORC1 mutants (i.e. Psm1-S1022), while others (i.e. Mis4-183) may behave differently.

It is clear that there are phenotypic differences between the suppression of mis4-ts by rapamycin treatment or by genetic alteration of TORC1. This can be seen also in our ChIP analysis of Rad21 distribution at CARs. The trend is upward, but the pattern is not identical. We have added the following text to summarize the above considerations:

“It is important to note at this stage that, although rapamycin and TORC1 mutants both decrease TORC1 kinase activity, the two are not equivalent. The mechanisms by which TORC1 kinase activity is reduced are different, and TORC1 mutants suppress the mis4G1487D phenotype more effectively than rapamycin. It is known that bona fide TORC1 substrates respond differently to rapamycin. Some phosphosites show acute sensitivity, while others are less sensitive or even insensitive (Kang et al, 2013). TORC1 mutants cause a constitutive decrease in activity with possible adaptive effects, whereas rapamycin is applied for a single cell cycle. While the lists of affected TORC1 substrates may overlap, they are unlikely to be identical. Furthermore, the phosphorylation level of the relevant substrates is not necessarily altered to the same extent. It is therefore remarkable that negative regulation of TORC1 by rapamycin or a genetic mutation both alleviate mis4G14878D phenotypes and have a fairly similar effect on cohesin dynamics.”

(2) The phosphorylation sites identified on cohesin subunits do not match known AGC kinase consensus motifs, raising the possibility that the modifications are indirect.

The genetic and biochemical analyses provided in this study show that the AGC kinases Sck1 and Sck2 influence cohesin phosphorylation and function. Whether Sck1, Sck2 or TORC1 directly phosphorylates cohesin components are the next questions to address. The fact that the phosphorylation of Psm1-S1022 and Mis4-S183 were never abolished in the sck1-2 mutants may suggest they are indirectly involved. This should be taken with caution because we have been using deletion mutants. In this situation, cells adapt and other kinases may substitute, at least partially (Plank et al, 2020, PMID: 32102971). Asking whether cohesin components display consensus sites for AGC kinases is a complementary approach. The consensus site for Sck1 and Sck2 is unknown. If we assume some conservation with budding yeast SCH9, the consensus sequence would be RRxS/T. Psm1S1022 (DQMSP) and Mis4-S183 (QLCSP) do not fit the consensus. However, this kind of information should be taken with care as many SCH9-dependent phosphorylation sites did not fall within the consensus in a study using analogue-sensitive AGC kinases and phosphoproteomics (Plank et al, 2020, PMID: 32102971). Alternatively, Sck1-2 may regulate other kinases. Indeed Psm1-S1022 and Mis4-183 lie within CDK consensus sites and Psm1-S1022 phosphorylation is Pef1-dependent. In summary, yes, the changes may be indirect, that remains to be seen, but in any case they are influenced by TORC1 signalling. The following paragraph was added:

“The consensus site for Sck1 and Sck2 is unknown. If we assume some conservation with budding yeast SCH9, the consensus sequence would be RRxS/T. Psm1-S1022 (DQMSP) and Mis4-S183 (QLCSP) do not fit the consensus. However, this should be taken with care as many SCH9-dependent phosphorylation sites did not fall within the consensus in a study using analogue-sensitive AGC kinases and phosphoproteomics (Plank et al, 2020). Alternatively, Sck1-2 may regulate other kinases. Indeed Psm1-S1022 and Mis4-183 lie within CDK consensus sites and Psm1-S1022 phosphorylation is Pef1-dependent.”

(3) The study relies heavily on one TORC1 mutant allele (mip1-R401G), and additional alleles could strengthen the generality of the findings.

It is true that we focused our attention on mip1-R401G, which is present in all the experiments presented. That said, other alleles were used in one or more figures. Five mip1 alleles and one tor2 allele were identified as mis4-ts suppressors (Fig. 1). We have also shown that another mip1 allele, mip1-Y533A, created by another group (Morozumi et al, 2021), is also a suppressor of mis4-ts and affects the phosphorylation of Mis4-S183 and Psm1-S1022 (Fig. 1, Figure 5—figure supplement 1). To this we can add the effect of mutants that render TORC1 hyperactive (Fig. 1E, Fig. 2H) as well as AGC kinase mutants (Figure 5—figure supplement 3.). And finally, the effect of rapamycin. So yes, mip1-R401G has been used extensively, but we have still broadly covered the TORC1 signalling pathway.

(4) Furthermore, while the results suggest that nutrient availability influences cohesin function, this is not directly tested by comparing growth or cohesin dynamics under defined nutrient conditions

We agree that studying the dynamics of cohesin, genome folding and gene expression in relation to nutrient availability is a very exciting topic, and we hope to address these issues in detail in the future.

**Reviewer #2 (Public review):**
Summary:In this study, the authors follow up on a previous suppressor screen of a temperaturesensitive allele of mis4 (mis4-G1487D), the cohesin loading factor in *S. pombe*, and identify additional suppressor alleles tied to the *S. pombe* TORC1 complex. Their analysis suggests that these suppressor mutations attenuate TORC1 activity, while enhanced TORC1 activity is deleterious in this context. Suppression of TORC1 activity also ameliorates chromosome segregation and spindle defects observed in the mis4-G1487D strain, although some more subtle effects are not reconstituted. The authors provide evidence that this genetic suppression is also tied to the reconstitution of cohesin loading. Moreover, disrupting TORC1 also enhances Mis4/cohesin association with chromatin (likely reflecting enhanced loading) in WT cells, while rapamycin treatment can enhance the robustness of chromosome transmission. These effects likely arise directly through TORC1 or its downstream effector kinases, as TORC1 co-purifies with Mis4 and Rad21; these factors are also phosphorylated in a TORC1-dependent fashion. Disrupting Sck2, a kinase downstream of TORC1, also suppresses the mis4-G1487D allele while simultaneous disruption of Sck1 and Sck2 enhances cohesin association with chromatin, albeit with differing effects on phosphorylation of Mis4 and Psm1/Scm1. Phosphomutants of Mis4 and Psm1 that mimic observed phosphorylation states identified by mass spectrometry that are TORC1-dependent also suppressed phenotypes observed in the mis4-G1487D background. Last, the authors provide evidence that the mis4-G1487D background and TORC1 mutant backgrounds display an overlap in the dysregulation of genes that respond to environmental conditions, particularly in genes tied to meiosis or other "stress".Overall, the authors provide compelling evidence from genetics, biochemistry, and cell biology to support a previously unknown mechanism by which nutrient sensing regulates cohesin loading with implications for the stress response. The technical approaches are generally sound, well-controlled, and comprehensive.Specific Points:(1) While the authors favor the model that the enhanced cohesin loading upon diminished TORC1 activity helps cells to survive harsh environmental conditions, as starvation of *S. pombe* also drives commitment to meiosis, it seems as plausible that enhanced cohesin loading is related to preparing the chromosomes to mate.(2) Related to Point 1, the lab of Sophie Martin previously published that phosphorylation of Mis4 characterizes a cluster of phosphotargets during starvation/meiotic induction (PMID: 39705284). This work should be cited, and the authors should interrogate how their observations do or do not relate to these prior observations (are these the same phosphosites?).

We agree this is a possibility and the following paragraph was added in the discussion section:

“TORC1-based regulation of cohesin may be relevant to preparing cells for meiosis. Since nitrogen deprivation stimulates meiosis initiation, subsequent TORC1 down-regulation may regulate the cohesin complex, preparing the chromosomes for fusion and meiosis. A recent phosphoproteomic study conducted by Sophie Martin's laboratory showed that Mis4-S107 phosphorylation increases during cellular fusion (Bérard et al, 2024). It is unknown whether the phosphorylation of S107 is controlled by TORC1 signalling. As the phosphorylation of Mis4-S183 and Psm1-S1022 was not detected in these experiments, the potential involvement of the TORC1-cohesin axis in the sexual programme remains to be investigated.”

(3) It would be useful for the authors to combine their experimental data sets to interrogate whether there is a relationship between the regions where gene expression is altered in the mis4-G1487D strain and changes in the loading of cohesin in their ChIP experiments.(4) Given that the genes that are affected are predominantly sub-telomeric while most genes are not affected in the mis4-G1487D strain, one possibility that the authors may wish to consider is that the regions that become dysregulated are tied to heterochromatic regions where Swi6/HP1 has been implicated in cohesin loading

We agree that it would be interesting to see if there are correlations between cohesin positioning, heterochromatin and gene expression. That said, this would need to be done at the whole-genome level and include many other parameters (genome folding, histone modifications, Pol2 occupancy). These issues require substantial investment and may be addressed in a follow-up project.

(5) It would be helpful to show individual data points from replicates in the bar graphs - it is not always clear what comprises the data sets, and superplots would be of great help.

We verified that the figure captions clearly indicate the data sets considered, their mean, standard deviation, and statistical analysis method. As for the type of plot, we used the tools at our disposal.

**Recommendations for the authors:**

**Reviewer #1 (Recommendations for the authors):**
Besson et al. investigate how the nutrient-responsive TORC1 signaling pathway modulates cohesin function in *S. pombe*. Using a genetic screen, the authors identify TORC1 mutants that suppress the thermosensitive growth defects of a cohesin loader mutant (mis4-G1487D). They show that reducing TORC1 activity-either genetically or pharmacologically-enhances cohesin binding to chromosomal sites (CARs), improves chromosome segregation, and alters the phosphorylation state of cohesin and its loader. They also show, through coimmunoprecipitation, that TORC1 and cohesin physically associate, and that this functional interaction extends to the transcriptional regulation of stress-responsive, subtelomeric genes. Together, the data suggest that environmental cues influence chromosome stability and gene expression via a TORC1-cohesin axis.Overall, the study is well-supported by thoughtful genetic epistasis analyses and a combination of genetic, biochemical, cell biological, and transcriptomic approaches. While not all data are equally strong, the cumulative evidence convincingly supports the authors' conclusions.Specific Concerns and Suggestions(1) Figure 2A - Division rates of wild-type and mip1-R401G cells are missing and should be provided for proper comparison.

This is now done in revised Figure 2A. We also made a change in the manuscript, replacing “The mip1-R401G mutation efficiently suppressed the proliferation and viability defects (Figure 2A)” by “The mip1-R401G mutation efficiently attenuated the proliferation and viability defects (Figure 2A)”, to acknowledge the fact that the proliferation rate did not return to wild-type levels.

(2) Figure 3 - Figure Supplement 1 - The authors claim that "Rapamycin treatment during a single cell cycle provoked a similar effect although less pronounced." However, for most CARs, the effect appears insignificant. This should be acknowledged in the text.

The text has been changed accordingly:

“Rapamycin treatment during a single cell cycle provoked a similar stimulation of Rad21 binding at CARs (Figure 3—figure supplement 1), albeit with noticeable differences. In mis4+ cells, both mip1-R401G and rapamycin induced a significant increase in Rad21 binding at several CARs (tRNA-left, cc2, 3323, NTS, Tel1-R). However, some CARs that exhibited increased Rad21 binding in the mip1 mutant did not respond significantly to rapamycin (dg2-R, tRNA-R). Conversely, rapamycin (but not mip1-R401G) induced a significant increase in Rad21 binding at imr2-L and CAR1806 (Figure 3D and Figure 3— figure supplement 1). In the mis4-G1487D mutant background, mip1-R401G induced a significant increase in Rad21 binding at all examined sites (Figure 3B). Similarly, rapamycin did increase Rad21 binding at all sites but only at the Tel1-R site did this reach statistical significance (Figure 3—figure supplement 1).”

(3) Figure 4 - The analysis of interactions between TORC1 and the cohesin complex is somewhat limited. The authors may wish to test interactions between Mip1 and cohesin subunits (e.g., Rad21). More interestingly, it would be valuable to explore whether MIP1 mutations that suppress cohesin mutants affect the interaction between Tor2 and Rad21.

We have added some additional data that answer this question (Figure 4—figure supplement 1) and a paragraph in the manuscript:

“Tor2, the kinase subunit of TORC1, is particularly well detected in Rad21 and Mis4 coimmunoprecipitation experiments (Figure 4 and Figure 4—figure supplement 1). To determine whether the R401G mutation in Mip1 affects these interactions, coimmunoprecipitation experiments were repeated in both the mip1-R401G and mip1+ contexts. The data obtained indicate that Tor2 co-immunoprecipitation with Mis4 and Rad21 is largely unaffected by the mip1-R401G mutation (Figure 4—figure supplement 1). If mip1-R401G affects the regulation of cohesin by TORC1, this does not appear to stem from a gross defect in their interaction, at least at this level of resolution.”

(4) Figure 5 - There appears to be a lack of correlation between cohesin subunit phosphorylation in TORC1-reducing mutants and in response to rapamycin. The reason for this discrepancy is unclear.

This point was addressed in the previous section (Public review, reviewer 1, point 1). The response is pasted below:

The basis of our study was to search for suppressor mutants, a situation in which an unviable strain becomes viable. It turns out that the suppressor mutants affect TORC1, necessarily in a partial manner given that TORC1 kinase activity is essential for proliferation. Likewise rapamycin partially inhibits TORC1 and does not prevent proliferation of wild-type *S. pombe* cells. TORC1 mutants cause a constitutive decrease in activity with possible adaptive effects, whereas rapamycin is applied for a single cell cycle. In addition, it is known that bona fide TORC1 substrates respond differently to rapamycin. Some phosphosites show acute sensitivity, while others are less sensitive or even insensitive (Kang et al., 2013, PMID: 23888043). Therefore, both hypomorphic TORC1 genetic mutants and rapamycin treatment result in partial inhibition of TORC1 kinase activity. While the lists of affected TORC1 substrates may overlap, they are unlikely to be identical. Furthermore, the phosphorylation level of the relevant substrates is not necessarily altered to the same extent. Nevertheless, both conditions suppress the heatsensitive phenotype of the mis4 mutant, although the suppressor effect of rapamycin is weaker. Consequently, some phosphorylation sites involved in mis4-ts suppression may behave similarly in rapamycin and TORC1 mutants (i.e. Psm1-S1022), while others (i.e. Mis4-183) may behave differently.

It is clear that there are phenotypic differences between the suppression of mis4-ts by rapamycin treatment or by genetic alteration of TORC1. This can be seen also in our ChIP analysis of Rad21 distribution at CARs. The trend is upward, but the pattern is not identical. We have added the following text to summarize the above considerations:

“It is important to note at this stage that, although rapamycin and TORC1 mutants both decrease TORC1 kinase activity, the two are not equivalent. The mechanisms by which TORC1 kinase activity is reduced are different, and TORC1 mutants suppress the mis4G1487D phenotype more effectively than rapamycin. It is known that bona fide TORC1 substrates respond differently to rapamycin. Some phosphosites show acute sensitivity, while others are less sensitive or even insensitive (Kang et al, 2013). TORC1 mutants cause a constitutive decrease in activity with possible adaptive effects, whereas rapamycin is applied for a single cell cycle. While the lists of affected TORC1 substrates may overlap, they are unlikely to be identical. Furthermore, the phosphorylation level of the relevant substrates is not necessarily altered to the same extent. It is therefore remarkable that negative regulation of TORC1 by rapamycin or a genetic mutation both alleviate mis4G14878D phenotypes and have a fairly similar effect on cohesin dynamics.”

(5) The phosphorylation sites examined on cohesin subunits are not canonical AGC kinase consensus motifs, suggesting they are unlikely to be direct targets of Sck1 or Sck2. I suggest that this point should be mentioned in the manuscript.

This is now done:

“The consensus site for Sck1 and Sck2 is unknown. If we assume some conservation with budding yeast SCH9, the consensus sequence would be RRxS/T. Psm1-S1022 (DQMSP) and Mis4-S183 (QLCSP) do not fit the consensus. However, this should be taken with care as many SCH9-dependent phosphorylation sites did not fall within the consensus in a study using analogue-sensitive AGC kinases and phosphoproteomics (Plank et al, 2020). Alternatively, Sck1-2 may regulate other kinases. Indeed Psm1-S1022 and Mis4-183 lie within CDK consensus sites and Psm1-S1022 phosphorylation is Pef1-dependent.”

(6) Figure 5 - Figure Supplement 3 - The reduction in Psm1 phosphorylation in the sck1Δ sck2Δ double mutant is not convincing without replicates and statistical analysis.

This is now done and the data are presented in Figure 5—figure supplement 3. Panel D shows the data for Psm1-S1022p and Panel E for Mis4-S183p. Each graph shows the mean ratios +/- SD from 3 experiments.

(7) Figure 5C - It would be helpful if the authors validated the effect of pef1 deletion on Mis4 phosphorylation by Western blotting, rather than relying solely on mass spectrometry data.

This is now done. The data appears in Figure 5—figure supplement 2, panel B.

(8) The statement: "The frequency of chromosome segregation defects of mis4‐G1487D was markedly reduced in a sck2‐deleted background and further decreased by the additional deletion of sck1 (Figure 5-figure supplement 3)" is not supported by the data. According to the figure, the difference between sck2Δ and sck1Δ sck2Δ is not statistically significant.

The sentence was changed to:

“The frequency of chromosome segregation defects in the mis4-G1487D strain remained unchanged in a sck1-deleted background, but was significantly reduced when either the sck2 or both the sck1 and sck2 genes were deleted (Figure 5—figure supplement 3).”

(9) Figure 6A - The data shown are not convincing. The double mutants carrying the phosphomimetic and phospho-null psm1 alleles should be shown on the same plate for direct comparison.

This is now done. The new data are shown Figure 6A.

(10) Figure 6E - The wild-type control is missing. Including it would provide an essential reference point to assess whether the mutants rescue cohesin binding to wild-type levels.

This is true that the effects were small when compared to wild-type but still significant when compared to mis4-G1487D. The comparison with wild-type is now available in Figure 6—figure supplement 1 and the paragraph was modified accordingly:

“Cohesin binding to CARs as assayed by ChIP tend to increase for the mutants mimicking the non-phosphorylated state and to decrease with the phospho-mimicking forms (Figure 6E). The rescue of mis4-G1487D by the non-phosphorylatable form was modest but significant, notably within centromeric regions (imr2-L, dg2-R) and at the telomere (Tel1-R) site (Figure 6E and see Figure 6—figure supplement 1 for comparison with wild-type levels). Conversely, the mutant mimicking the phosphorylated state displayed a significant reduction of Rad21 binding at those sites as well as to several other sites at the centromere (cc2, tRNA-R), CAR2898, and at the ribosomal non-transcribed spacer site NTS.”

Limitations of the Study (not requiring additional experiments for publication, but worth noting).

(11) The authors suggest that nutrient status affects cohesin, but this is not directly demonstrated-e.g., by comparing growth or cohesin dynamics or phosphorylation under defined nutrient conditions. That said, the paper is sufficiently detailed to allow this question to be addressed in follow-up work.

We agree that studying the dynamics of cohesin, genome folding and gene expression in relation to nutrient availability is a very exciting topic, and we hope to address these issues in detail in the future.

(12) The upstream signaling cascade remains unresolved. The identity of kinases downstream of TORC1 (e.g., whether Sck1/Sck2 or other factors are responsible) and whether TORC1 directly phosphorylates Mis4 or Psm1 are not established.

This is something we can all agree on, and it might be something we look at in a future project.

(13) The conclusions rely heavily on one TORC1 mutant allele (mip1-R401G). While this allele is informative, additional alleles or orthogonal methods could further support the generality of the findings.

It is true that we focused our attention on mip1-R401G, which is present in all the experiments presented. That said, other alleles were used in one or more figures. Five mip1 alleles and one tor2 allele were identified as mis4-ts suppressors (Fig. 1). We have also shown that another mip1 allele, mip1-Y533A, created by another group (Morozumi et al, 2021), is also a suppressor of mis4-ts and affects the phosphorylation of Mis4-S183 and Psm1-S1022 (Fig. 1, Figure 5—figure supplement 1). To this we can add the effect of mutants that render TORC1 hyperactive (Fig. 1E, Fig. 2H) as well as AGC kinase mutants (Figure 5—figure supplement 3.) and finally, the effect of a transient treatment with rapamycin. So yes, mip1-R401G has been used extensively, but we have still broadly covered the TORC1 signalling pathway.

**Reviewer #2 (Recommendations for the authors):**
(1) Given the lack of CTCF in fission yeast, it is worth noting that cohesin ChIP data nonetheless can predict topological domains, which reinforces its important role in dictating chromatin folding (PMID: 39543681).

We thank the reviewer for this suggestion. We now refer to this study in the discussion section.

(2) Providing context for the *S. pombe* nomenclature for the conserved cohesin subunits would help the reader navigate the manuscript, possibly using a cartoon as for the TORC complexes. For example, Psm1 (aka Smc1) is not introduced and therefore its phosphorylation comes into the manuscript without explanation.

Cohesin subunits and their names are given in the introduction section.